

# The CoralHydro2k Database: a global, actively curated compilation of coral δ18O and Sr/Ca proxy records of tropical ocean hydrology and temperature for the Common Era

Rachel M. Walter[1], Hussein R. Sayani[1†], Thomas Felis[2], Kim M. Cobb[1], Nerilie J. Abram[3,4], Ariella K. Arzey[5], Alyssa R. Atwood[6], Logan D. Brenner[7], Émilie P. Dassié[8], Kristine L. DeLong[9], Bethany Ellis[4,10], Matthew J. Fischer[11], Nathalie F. Goodkin[12], Jessica A. Hargreaves[3,4], K. Halimeda Kilbourne[13], Hedwig Krawczyk[14], Nicholas P. McKay[15], Sujata A. Murty[16], Riovie D. Ramos[17], Emma V. Reed[18], Dhrubajyoti Samanta[19], Sara C. Sanchez[20], Jens Zinke[14], and PAGES CoralHydro2k Project Members[+]

[1]School of Earth and Atmospheric Sciences, Georgia Institute of Technology, Atlanta, 30332, USA
[†]Presently at the Department of Earth, Ocean and Atmospheric Science, Florida State University, Tallahassee, 32306, FL, USA
[2]MARUM - Center for Marine Environmental Sciences, University of Bremen, 28359 Bremen, Germany
[3]ARC Centre of Excellence for Climate Extremes, The Australian National University, Canberra, 2601, Australia.
[4]Research School of Earth Sciences, The Australian National University, Canberra, 2601, Australia
[5]School of Earth, Atmospheric and Life Sciences, University of Wollongong, Wollongong, 2522, Australia
[6]Department of Earth, Ocean and Atmospheric Science, Florida State University, Tallahassee, 32306, USA
[7]Department of Environmental Science, Barnard College, New York, 10027, USA
[8]UMR 5805 EPOC - CNRS - OASU - Université de Bordeaux, Pessac, 33615, France
[9]Department of Geography and Anthropology and Coastal Studies Institute, Louisiana State University, Baton Rouge, 70803, USA
[10]New South Wales Department of Primary Industries, Orange, 2800, Australia
[11]NST Environment, ANSTO, Lucas Heights, 2234, Australia
[12]Department of Earth and Planetary Sciences, American Museum of Natural History, New York, 10021, USA
[13]Chesapeake Biological Laboratory, University of Maryland Center for Environmental Science, Solomons, 20657, USA
[14]School of Geography, Geology and the Environment, University of Leicester, Leicester, LE1 7RH, UK
[15]School of Earth and Sustainability, Northern Arizona University, Flagstaff, 86011, USA
[16]Department of Atmospheric and Environmental Sciences, University at Albany, State University of New York, Albany, 12222, USA
[17]Department of Environmental Science, William Paterson University of New Jersey, Wayne, 07470, USA
[18]Department of Geosciences, University of Arizona, Tucson, 85721, USA
[19]Earth Observatory of Singapore, Nanyang Technological University, Singapore, 639798, Singapore
[20]Department of Atmospheric and Oceanic Sciences, University of Colorado, Boulder, 80309, USA
[+]A full list of authors appears at the end of the paper

*Correspondence to*: Hussein R. Sayani (hsayani@fsu.edu)

**Abstract.** The response of the hydrological cycle to anthropogenic climate change, especially across the tropical oceans, remains poorly understood due to the scarcity of long instrumental temperature and hydrological records. Massive shallow-water corals are ideally suited to reconstructing past oceanic variability as they are widely distributed across the tropics, rapidly deposit calcium carbonate skeletons that continuously record ambient environmental conditions, and can be sampled at monthly to annual resolution. Most coral-based reconstructions utilize stable oxygen isotope composition ($\delta^{18}O$) that tracks the combined change in sea surface temperature (SST) and the oxygen isotopic composition of seawater ($\delta^{18}O_{sw}$), a measure of hydrologic variability. Increasingly, coral $\delta^{18}O$ time series are paired with time series of strontium-to-calcium ratios (Sr/Ca), a proxy for SST, from the same coral to quantify temperature and $\delta^{18}O_{sw}$ variability through time. To increase



the utility of such reconstructions, we present the CoralHydro2k database: a compilation of published, peer-reviewed coral Sr/Ca and $\delta^{18}$O records from the Common Era. The database contains 54 paired Sr/Ca-$\delta^{18}$O records and 125 unpaired Sr/Ca or $\delta^{18}$O records, with 88% of these records providing data coverage from 1800 CE to present. A quality-controlled set of metadata with standardized vocabulary and units accompanies each record, informing the use of the database. The CoralHydro2k database tracks large-scale temperature and hydrological variability. As such, it is well-suited for investigations of past climate variability, comparisons with climate model simulations including isotope-enabled models – and application in paleo-data assimilation projects. The CoralHydro2k database will be available on the NOAA National Center for Environmental Information's Paleoclimate data service with serializations in MATLAB, R, Python, and LiPD.

## 1 Introduction

The global hydrological cycle is changing in response to ongoing anthropogenic climate change (Held and Soden, 2006; Cheng et al., 2020), yet regional trends in hydrology remain uncertain in many areas of the world (Song et al., 2021; Madakumbura et al., 2021; Ummenhofer et al., 2021). Observed and projected trends in large-scale hydrology are consistent with the "wet get wetter, dry get drier" paradigm (Held and Soden, 2006) as surface ocean fluxes increase as the planet warms. Rising global temperatures means that the atmosphere can hold more moisture, which contributes to more extreme rainfall across a variety of spatiotemporal scales. In the tropics, many aspects of large-scale hydrology are tied to changes in large-scale coupled ocean-atmosphere dynamics associated with the El Niño-Southern Oscillation (ENSO; Power et al., 2013; Cai et al., 2014), tropical Pacific decadal variability (Gu and Adler, 2013; Dong and Dai, 2015), the Indian Ocean Dipole (Webster et al., 1999; Saji et al., 1999; Cai et al., 2019), and Atlantic Multidecadal Variability (Zhang et al., 2019), to name a few of the most prominent modes.

The detection of potential anthropogenic trends in regional hydrology against a rich background of natural regional hydrological variability is complicated by a dearth of instrumental climate data from across the tropics. In particular, instrumental sea surface temperature (SST) observations are sparse prior to the advent of satellites in 1979 (Reynolds et al., 2002; Rayner et al., 2003; Freeman et al., 2017; Huang et al., 2017; Kennedy et al., 2019) and the vast majority of sea surface salinity (SSS) observations only become available in the 1990's, with the advent of the Global Tropical Moored Buoy array (McPhaden et al., 1998, 2010) and World Ocean Circulation Experiment (WOCE) (Good et al., 2013; Friedman et al., 2017; Cheng et al., 2020; Gould and Cunningham, 2021). Both natural and anthropogenic shifts in regional hydroclimate on interannual to multi-decadal timescales have profound impacts on societies, economies, and ecosystems, such that resolving regional trends in past hydrological variability prior to available observational records is a scientific and societal priority.

Shallow-water corals have been extensively used to reconstruct past regional to oceanic-scale climate variability at data scarce locations in the tropical and subtropical oceans (as reviewed by Gagan et al., 2000; Corrège, 2006; Lough, 2010; Felis, 2020). Seasonally banded coral skeletons (e.g., Lough and Barnes, 1997) can yield monthly to annually resolved proxy records that can be calibrated to instrumental climate observations and thus used to extend the relatively short instrumental SST and SSS records back to the pre-instrumental era. Most coral-based reconstructions are based on the oxygen isotopic composition ($\delta^{18}$O) and/or strontium-to-calcium ratios (Sr/Ca) of coral skeletal aragonite. Coral $\delta^{18}$O tracks changes in SST as well as the oxygen isotopic composition of seawater ($\delta^{18}O_{sw}$) (Epstein et al., 1953; Weber and Woodhead, 1972). Like salinity,



variability in $\delta^{18}O_{sw}$ reflects the balance of precipitation and evaporation, terrestrial runoff, continental ice melt and formation, and ocean circulation and mixing (e.g., LeGrande and Schmidt, 2006, 2011; Hasson et al., 2013; Conroy et al., 2014). Coral

Sr/Ca primarily tracks SST variability (Weber, 1973; Smith et al., 1979; Beck et al., 1992) and can be used to decouple the temperature and $\delta^{18}O_{sw}$ signals in coral $\delta^{18}O$ records (e.g., Gagan et al., 1998; Ren et al., 2003; Corrège, 2006; Cahyarini et al., 2008). As such, paired coral $\delta^{18}O$ and Sr/Ca records can be used to independently investigate trends in SST and hydrology (Hendy et al., 2002; Linsley et al., 2006; Quinn et al., 2006; Zinke et al., 2008; Felis et al., 2009, 2018; Hetzinger et al., 2010; Nurhati et al., 2011; Cahyarini et al., 2014; Wu et al., 2014; Murty et al., 2017, 2018b; Hennekam et al., 2018; von Reumont

et al., 2018; Pfeiffer et al., 2019; Ramos et al., 2019, 2020; Sayani et al., 2019). Whereas coral-based reconstructions have provided much-needed insights on local SST and SSS at many tropical sites, the utility of this archive in reconstructing regional- and global-scale signals has been limited by the scarcity of long-term paired coral $\delta^{18}O$ and Sr/Ca records and the methodological challenges of deriving seawater $\delta^{18}O$ changes from these records.

Recent data synthesis efforts within the international paleoclimate community, under the auspices of the Past Global

Changes (PAGES) 2k Network, have produced several databases to contextualize modern climate change against the background of natural climate variability over the last ~2000 years; a time interval known as the Common Era (CE) (e.g., PAGES 2k Consortium, 2013; Tierney et al., 2015; PAGES2k Consortium, 2017; Konecky et al., 2020). These data sets, combined with climate simulations, have been instrumental in improving our understanding of CE climate variability and its dynamics (e.g., Abram et al., 2016; Neukom et al., 2019; PAGES 2k Consortium, 2019). Notably, the PAGES Ocean2k project

compiled a network of published coral $\delta^{18}O$, Sr/Ca, and extension rate records to reconstruct tropical SST evolution over the past few centuries (Tierney et al., 2015). More recently, the PAGES Iso2k project compiled water isotope records from a variety of terrestrial and marine archives (Konecky et al., 2020), including corals, to investigate temperature-driven changes in the global hydrological cycle (Konecky et al., submitted). Building on these previous efforts, the CoralHydro2k project brought the global coral paleoclimate community together to address existing data archiving needs and access issues as well

as the lack of standardized, best-practice methodology for calibrating coral proxies to climate variables and deriving $\delta^{18}O_{sw}$ changes from paired $\delta^{18}O$ and Sr/Ca records.

Here we present the PAGES CoralHydro2k database: a new, actively curated compilation of coral $\delta^{18}O$ and Sr/Ca records from the last 2,000 years that serve as proxies for near-surface conditions across the tropical and subtropical oceans. This new database employs metadata standards established by Marine Annually Resolved Proxy Archives (MARPA, Dassié

et al., 2017) and Paleoclimate Community reporTing Standard (PaCTS 1.0, Khider et al., 2019), and is built using the Linked Paleo Data (LiPD) framework (McKay and Emile-Geay, 2016). This first paper from the CoralHydro2k project outlines this new database, its functionality, as well as plans for active curation of records and future updates. As this database represents the most comprehensive collection of coral records to date, we highlight the existing spatiotemporal coverage and identify opportunities for future data collection.



## 2 Methods

### 2.1 Collaborative model

CoralHydro2k is one of nine projects that make up Phase 3 of the PAGES 2k Network, a long-standing effort to study climate variability over the last 2,000 years (PAGES 2k Network Coordinators, 2017). The CoralHydro2k project was established at the 2017 PAGES Open Science Meeting in Zaragoza, Spain, inspired by the PAGES Hydro2k Workshop in 2016 (PAGES Hydro2k Consortium, 2017). Recurring calls for participation were distributed within the international paleoclimate community to recruit a team with diverse expertise ranging from coral paleothermometry to paleodata assimilation. The resulting CoralHydro2k community is composed of 40+ volunteer scientists from all academic levels, including undergraduate and graduate students, postdoctoral researchers, and early to senior-level scientists from a variety of international academic and research institutions. Data compilation, initial analysis, and interpretation were done collaboratively and subdivided among thematic working groups as the project progressed. The majority of the work was completed remotely and asynchronously across several virtual platforms (Google Suite, Slack, and Zoom). One in-person meeting with limited remote participation took place in 2019 as a side meeting at the 13th International Conference on Paleoceanography (ICP13) in Sydney, Australia (Hargreaves et al., 2020).

### 2.2 Record selection and aggregation

Record selection criteria for the CoralHydro2k database were designed to be as inclusive and comprehensive as possible to develop a versatile database that supports the project's goal of reconstructing tropical hydroclimatic variability at seasonal and longer timescales. The database also supports the broader climate community's need for a uniform global database of coral records for comparison with climate model output over the past 2000 years, especially isotope-enabled models. The CoralHydro2k team selected Common Era coral records that were at least 10 years in length; measured either $\delta^{18}O$, Sr/Ca, or both; were published in a peer-reviewed scientific journal; and were archived with an absolute chronology (i.e., time in years CE). For studies where "composite records", or average time series of multiple cores from a single site, were publicly available, we included either the composite record or its constituent time series but not both. Composite records are flagged as such in the database.

Coral records were sourced from past PAGES 2k data compilations with more restrictive selection criteria, such as Ocean2k (Tierney et al., 2015) and Iso2k (Konecky et al., 2020), as well as from public repositories such as the World Data Center PANGAEA (https://www.pangaea.de/) and the NOAA National Centers for Environmental Information (NCEI) World Data Service for Paleoclimatology (https://www.ncei.noaa.gov/products/paleoclimatology). For a few studies where data were not archived in public repositories, we retrieved the records from publications and supplemental information or contacted the corresponding authors. In addition to being compiled in the CoralHydro2k database, 27 previously unarchived records were submitted to NOAA's NCEI database for archival by CoralHydro2k project members.



## 2.3 Database organization

Coral records in the database are organized into seven groups based on the availability of paired proxy time series, temporal coverage, and record resolution (Table 1). Groups 1–3 contain records with paired Sr/Ca-$\delta^{18}$O time series. Group 1 records have monthly to bimonthly temporal resolution and cover at least 80% of the 20th century. Records in Group 2 are similar in resolution to records in Group 1, but cover less than 80% of the 20th century. Group 3 records contain any paired Sr/Ca-$\delta^{18}$O time series that have lower than bimonthly resolution. Group 4 records are $\delta^{18}$O-only time series with monthly to bimonthly resolution, while Group 5 records are $\delta^{18}$O-only time series with lower than bimonthly resolution. Groups 6 and 7 mirror Groups 4 and 5 respectively, but for Sr/Ca-only records.

**Table 1. Summary table of group descriptions for the CoralHydro2k database.**

| Group | Proxy data | Temporal resolution | Temporal coverage | # Records |
|---|---|---|---|---|
| 1 | paired Sr/Ca-$\delta^{18}$O | monthly to bimonthly | > 80 years of 20th century | 20 |
| 2 | paired Sr/Ca-$\delta^{18}$O | monthly to bimonthly | < 80 years of 20th century | 24 |
| 3 | paired Sr/Ca-$\delta^{18}$O | seasonal or lower | within CE | 10 |
| 4 | $\delta^{18}$O | monthly to bimonthly | within CE | 56 |
| 5 | $\delta^{18}$O | seasonal or lower | within CE | 23 |
| 6 | Sr/Ca | monthly to bimonthly | within CE | 36 |
| 7 | Sr/Ca | seasonal or lower | within CE | 10 |

Following the Iso2k database protocols (Konecky et al., 2020), each record in the CoralHydro2k database is assigned a unique nine-digit alphanumeric identifier. These unique identifiers are generated using the first two letters of the lead author surname (AN), the last two digits of publication year (01), a three-letter code indicating the location of the record (ABC), and a two-digit core-ID number (02). The two-digit core-ID number begins at '01' by default and increases with each successive record from the same site and publication. Identifiers have the final format "AN01ABC02". For example, record AB08MEN01 was published by Abram et al., in 2008, is a record from the Mentawai Islands, and is the first core from that study.

## 2.4 Metadata

The CoralHydro2k database contains 55 metadata fields that inform the use of each coral record: 32 metadata fields are standardized and quality controlled, while 23 fields are unstructured. Standardized metadata fields use controlled vocabulary or numeric information with uniform units, making them easily searchable by database users. Unstructured metadata are free-form text entries that are less rigorously quality controlled but are included to aid the interpretation of the coral records.

Metadata included in the CoralHydro2k database is organized into four categories (Entity, Publication, Analysis, and Calibration) based on standards recommended by MARPA (Dassié et al., 2017) and PaCTS1.0 (Khider et al., 2019). Entity



metadata provides identifying information for each coral record (Table 2), including geographic coordinates, location names, water depth of the coral colony, coral species, and any core names included in the original publications. Also included in entity metadata is resolution information and the start and end years of each record. Record resolution is provided as the minimum, maximum, mean, and median data points per year for each record. A nominal label for resolution (*monthly, bimonthly, quarterly, biannual, annual, or >annual*; described in Table 3), based on the modal resolution of a record, is also included to

allow users to easily search for records. The term '*_uneven*' is appended to the nominal label for records that have a variable resolution. Care should be used when interpolating these records to even sampling resolution for analysis because although most are relatively evenly sampled, some records have sections of substantially higher or lower resolution.

**Table 2. Entity metadata. Describes information relating directly to the coral proxy record, including location, core names, species, and time span. Standardized fields are italicized.**

| Field name | Variable | Type | Description |
|---|---|---|---|
| paleoData_ch2kCoreCode | Core ID | text | Core ID used to identify the record within the CoralHydro2k database. |
| paleoData_coralHydro2kGroup | Group | numeric | Group into which the record is sorted in the CoralHydro2k database, ranges from 1-7 based on criteria outlined in Table 1. |
| geo_latitude | Latitude | numeric | Latitude for the coral core. Positive values are north of the equator; negative values are south. |
| geo_longitude | Longitude | numeric | Longitude for the coral core. Positive values are east of the Prime Meridian; negative values are west. |
| geo_siteName | Site | text | Standardized location names. Names follow the format [island/city/province 1], [island/city/province 2 (optional)], [country]. Exceptions to this are reefs (reef, country) and other named, water-based locations (e.g. named areas within the Red Sea). |
| geo_secondarySiteName | Site 2 | text | Secondary location names. May include regional names (e.g. Line Islands, Great Barrier Reef) or names of specific sites (e.g. Silabu). |
| geo_ocean | Ocean basin | text | Ocean basin of the coral core as determined by its latitude and longitude according to the World Ocean Atlas (Boyer et al. 2018). |



| geo_ocean2 | Ocean basin 2 | text | Secondary ocean basin names listed in publication that are not included in the official World Ocean Atlas designations. |
|---|---|---|---|
| geo_elevation | Elevation | numeric | Elevation of corals. Values are negative to indicate corals were found below sea level. All elevation is expressed in meters (m). |
| paleoData_coreName | Core name | text | Core name as specified in publications and data sets. Allows for the tracing of the coral record through past and future publications. |
| paleoData_archiveSpecies | Coral species | text | Genus and species (if known) of the coral archive. Records where species name is unknown or not given are notated as '[Genus] sp.' |
| geo_description | Site Type | text | Any general description of the type of site in which the coral was found (e.g. fringing reef, open ocean, etc.). |
| hasResolution_nominal | Nominal resolution | text | Nominal temporal resolution of the proxy record. See Table 3 for term definitions. |
| hasResolution_hasMaxValue | Maximum resolution | numeric | Minimum temporal resolution of the proxy record. Units: years. |
| hasResolution_hasMeanValue | Mean resolution | numeric | Mean temporal resolution of the proxy record. Units: years. |
| hasResolution_hasMedianValue | Median resolution | numeric | Median temporal resolution of the proxy record. Units: years. |
| hasResolution_hasMinValue | Minimum resolution | numeric | Minimum temporal resolution of the proxy record. Units: years. |
| minYear | Minimum year | numeric | Minimum year of the proxy record. Expressed in integer years CE. |
| maxYear | Maximum year | numeric | Maximum year of the proxy record. Expressed in integer years CE. |
| paleoData_variableName | Data type | text | Data type for paleoData_values. Proxy types will be $\delta^{18}O$ (d18O), Sr/Ca (SrCa), or seawater $\delta^{18}O$ (d18Osw). Annual averages will have 'annual' |



| | | | appended to the proxy type, and error data will have 'Uncertainty' appended to the proxy type. |
|---|---|---|---|
| paleoData_values | Data | numeric | An Nx1 vector of proxy or error data. Data type is specified by paleoData_variableName. |
| paleoData_units | Data units | text | Units for paleoData_values. |
| year | Year | numeric | Time data for the proxy record in paleoData_values. |
| yearUnits | Year units | text | Units for year. |
| paleoData_TSid | TSid | text | Contains a unique LiPD ID string for the data within the database. Used to match error vectors with their given data vectors. |
| paleoData_hasUncertainty | Error TSid | text | Field containing the paleoData_TSid of the error timeseries for the given data set (error will be in paleoData_values of that TSid). |
| paleoData_isComposite | Composite data flag | logic | Indicates whether the proxy record in paleoData_values is a composite of multiple cores' proxy data. |
| paleoData_isAnomaly | Anomaly data flag | logic | Indicates whether proxy data in paleoData_values is anomaly data. |

**Table 3. Nominal resolution descriptors. Lists the definitions in 'data points per year' that were used to determine the nominal resolution label for each proxy record.**

| Nominal resolution | Data points per year |
|---|---|
| *monthly*; *monthly_uneven* | 12 data points per year; "_uneven" is added to records with variable resolutions that typically have over 12 data points per year |
| *bimonthly*; *bimonthly_uneven* | 6 data points per year; "_uneven" is added to records with variable resolutions that typically have 6–11 data points per year |
| *quarterly*; *quarterly_uneven* | 4 data points per year; "_uneven" is added to records with variable resolutions that typically have 4–5 data points per year |
| *biannual*; *biannual_uneven* | 2 data points per year; "_uneven" is added to records with variable resolutions that typically have 2–3 data points per year |
| *annual; annual_uneven* | 1 data point per year; "_uneven" is added to records with variable resolutions that typically have 1 data point per year |
| *>annual* | Less than 1 data point per year |



Publication metadata (Table 4) contains bibliographical information for each coral record including digital object identifiers (DOIs) for publications and links to the public repository from which the data was retrieved. For records featured
in multiple publications, bibliographical information for publications is stored in the order established by the source data repository. First citations are found in the *pub1* metadata fields, and subsequent citations are found in *pub2* and *pub3*.

Analysis metadata (Table 5) provides information about the laboratory analysis of the samples, including (when available) information related to subsampling the cores, coral extension rate and tissue thickness, the units of reported variables, and analytical precision for geochemical time series. When available, information on the measurement of the
international coral reference material JCp-1 (Okai et al., 2002; Hathorne et al., 2013) is included for Sr/Ca records. Calibration metadata (Table 6) includes any proxy-SST slopes, intercepts, correlations, and information about regression methods used, as reported in the original publications. These calibration metadata may differ from the standardized calibration results that we calculate across the whole database and report in section 3.2 below.

**Table 4. Publication metadata. Details publication information for up to three publications associated with each coral record.**
**Standardized fields are italicized.**

| Field name | Variable | Type | Description |
|---|---|---|---|
| *First author (publication X)* | pubX_firstauthor | text | First author listed for each listed publication (X = 1,2,3) |
| *Publication year (publication X)* | pubX_year | numeric | Year of publication for each listed publication (X = 1,2,3) |
| *DOI (publication X)* | pubX_doi | text | Digital object identifier (DOI) for each listed publication (X = 1,2,3) |
| Full citation (publication X) | pubX_citation | text | Complete citation for each listed publication (X = 1,2,3) |
| Title (publication X) | pubX_title | text | Title of each listed publication (X = 1,2,3) |
| Full author list (publication X) | pubX_author | text | Full list of authors from each listed publication (X = 1,2,3) |
| Journal (publication X) | pubX_journal | text | Journal of each listed publication (X = 1,2,3) |
| Original data source | originalDataUrl | text | Link to data set published in this database. |
| Additional data source | additionalDataUrl | text | Any additional links to published data related to this record. |

**Table 5. Analysis metadata. Coral sampling information, units used, and any additional notes on the coral record. Standardized fields are italicized.**

| Field name | Variable | Type | Description |
|---|---|---|---|
| paleoData_samplingResolution | Sampling resolution | text | Physical distance between individual samples from the coral archive. |

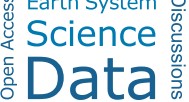

| | | | |
|---|---|---|---|
| paleoData_samplingNotes | Sampling notes | text | Any notes on sampling methods - point vs. continuous measurements, homogenization, etc. |
| *paleoData_coralExtensionRate* | Extension rate (mm/year) | numeric | Average coral extension rate in mm/year. If a range is given in the publication, 'Extension rate' is the average of the range. |
| paleoData_coralExtensionRateNotes | Extension rate notes | text | Average coral extension rate given in the publication. This entry includes any units, uncertainty, or ranges in values noted in publication. |
| *paleoData_coralTissueThickness* | Tissue thickness (mm) | numeric | Average coral tissue thickness in mm. If a range is given in the publication, 'Tissue thickness' is the average of the range. |
| *paleoData_jcpUsed* | JCP use flag | logic | Indicates whether the JCp-1 trace-element standard was used in the study (Okai et al. 2002, Hathorne et al. 2013). |
| *paleoData_jcpMeasured* | JCP value | numeric | If JCp-1 was used in the study, this is the measured value reported in the publication. Units are mmol/mol. |
| *paleoData_jcpCorrected* | JCP corrected | logic | Indicates whether proxy data in the study was standardized to JCp-1. |
| paleoData_jcpNotes | JCP notes | text | Any additional notes on information pertaining to JCp-1. |
| *paleoData_analyticalError* | Analytical error | numeric | Published analytic error for measured proxy values. |
| *paleoData_analyticalErrorUnits* | Analytical error units | text | Units for analytic error. |
| paleoData_notes | Additional coral record notes | text | Any notes on metadata, published values, citations, or the proxy record that did not fit in other fields. |



**Table 6. Calibration metadata. Any published information on the calibration of the coral record to sea surface temperature. Standardized fields are italicized.**

| Field name | Variable | Type | Description |
|---|---|---|---|
| calibration_method | Regression method | text | Regression method used with this data set in publication. Abbreviations are used for Ordinary Least Squares (OLS), Reduced Major Axis (RMA), Geometric Mean (GM), Weighted Least Squares (WLS), Multiple Linear Regression (MLR), and Composite Plus Scale (CPS). |
| calibration_dataset | SST product | text | SST data set used in publication for proxy-SST calibrations. |
| calibration_datasetRange | SST range | text | Average arithmetic SST range reported in publication for the coral site. Units: °C. |
| calibration_equationSlope | Proxy-SST slope | text | The published proxy-SST calibration slope for the coral record. Calibration equations take the form proxy = slope*SST + intercept. (Units: [paleoData_units]/°C ) |
| calibration_equationIntercept | Proxy-SST intercept | text | The published proxy-SST calibration intercept for the coral record. Calibration equations take the form proxy = slope*SST + intercept. (Units: [paleoData_units]/°C ) |
| calibration_equationR2 | Proxy-SST r-square value | text | The published proxy-SST calibration r-squared value for the coral record. |
| calibration_equationSlopeUncertainty | Proxy-SST slope uncertainty | text | Published proxy-SST slope uncertainty for the coral record. Calibration equations take the form proxy = slope*SST + intercept. (Units: [paleoData_units]/°C ) |



## 2.5 Quality control and validation

As records included in the CoralHydro2k database are published in peer-reviewed scientific journals, our quality control efforts were focused on the consistency of metadata and the accurate integration of records into the database. More specifically, the quality control team worked to ensure that (i) metadata and proxy time series were entered correctly into the database, (ii) metadata followed a standardized vocabulary or format, and (iii) records were sorted into the correct group based on the types of proxies available, length, and resolution. For sites where coral records were either extended or revised in subsequent studies, we include the most recent version of the record in the database and include citation information and other metadata from previous studies. A quality control checklist was used to ensure each field was in a standard format and contained information consistent with that in original publications and other online repositories. When information was unavailable, the corresponding fields were left blank.

Users of the database should not view the inclusion of a record as an endorsement of its fidelity by CoralHydro2k for reconstructing a climate parameter, as non-climatic factors (e.g., coral skeletal structure or growth rate) can complicate the extraction of climate signals from geochemical records (see Reed et al., 2021; DeLong et al., 2013, 2016). We strongly suggest users further assess records and original publications, or consult the original author or a coral paleoclimate expert if they have questions or concerns.

## 2.6 Relation to other PAGES2k products

CoralHydro2k was inspired by PAGES (2k) compilations of marine and hydrological proxy records such as Ocean2k (Tierney et al., 2015; McGregor et al., 2015), SISAL (Atsawawaranunt et al., 2018; Comas-Bru et al., 2020), and Iso2k (Konecky et al., 2020), but was created to address a different set of research questions. As the database is designed specifically for coral-based proxy records, we employ more inclusive record selection criteria that allow us to include records that do not meet the length requirements of previous PAGES 2k data compilations but are important to contextualize ongoing climate change during the Common Era. The CoralHydro2k database also contains new, updated, or extended records that were published after previous PAGES efforts, and will continue to be actively curated and updated annually. With a more comprehensive coral database, the CoralHydro2k project will investigate methodological differences in proxy-SST calibrations, explore methodologies for deriving coral-based $\delta^{18}O_{sw}$ reconstructions, refine proxy system models for coral Sr/Ca and $\delta^{18}O$ time series that enable proxy data and climate model intercomparison, and provide a denser proxy network for paleodata assimilation efforts.



# 3 Key characteristics of the CoralHydro2k database

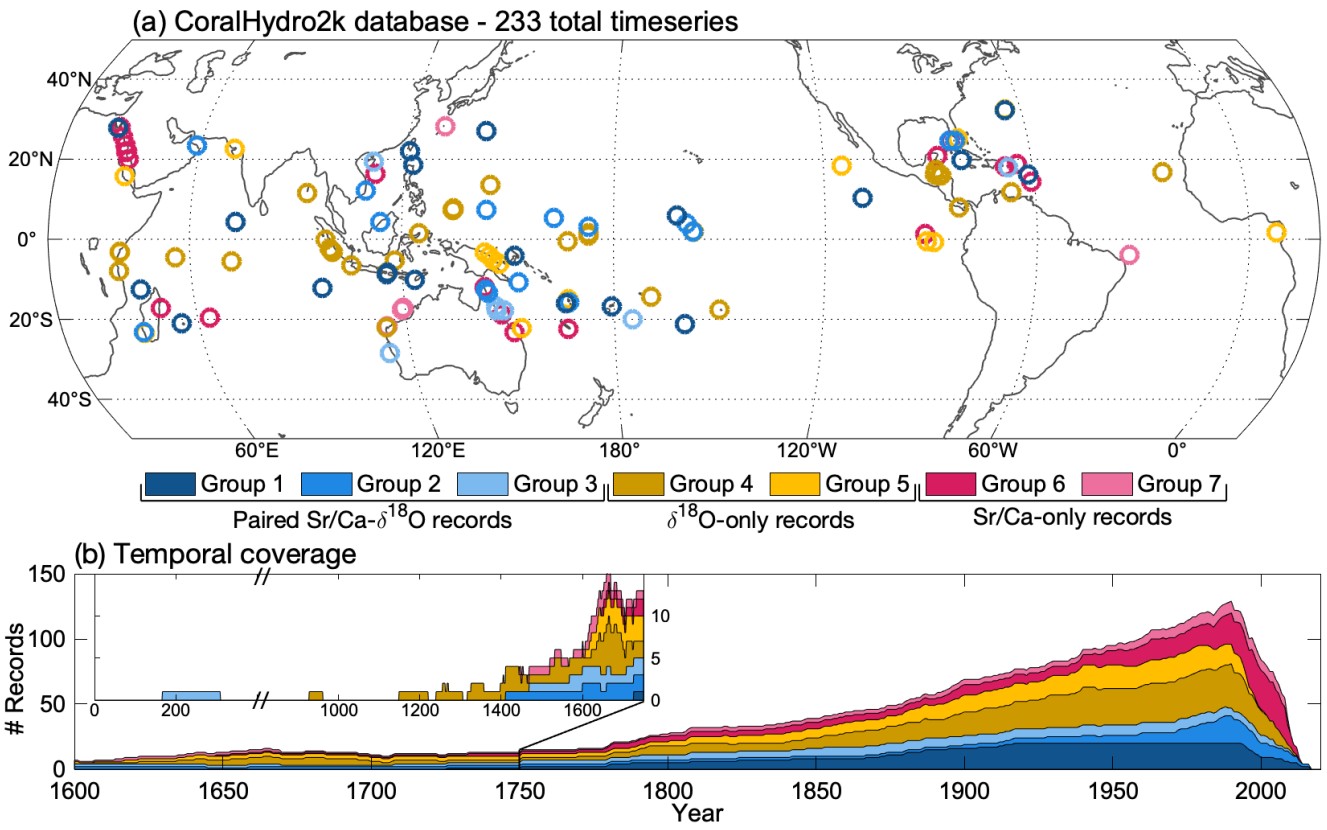

**Figure 1. CoralHydro2k database records are divided across Groups 1–7 based on their available proxy information. (a) Spatial distribution of all records in the CoralHydro2k database. (b) Temporal coverage of all records in the CoralHydro2k database. Inset shows earlier records (0–1750 CE).**

## 3.1 Spatial and temporal coverage

The CoralHydro2k database includes 233 proxy time series from 124 unique locations sorted into seven groups (Fig. 1a). The proxy time series are stored as "records", with 54 records containing paired Sr/Ca and $\delta^{18}$O time series, 79 records containing only $\delta^{18}$O time series, and 46 records containing only Sr/Ca time series. For 19 of the paired $\delta^{18}$O and Sr/Ca records, we also include in the database the coral-derived $\delta^{18}$O$_{sw}$ time series calculated by the authors of the original publication. Records in the CoralHydro2k database extend from 33° N to 28° S and across all tropical oceans. The majority of these records

are concentrated in the Indo-Pacific Warm Pool and the western tropical Atlantic, as conditions there are favorable for coral growth and reefs are more accessible to researchers. Record density is low in the eastern tropical Pacific and eastern tropical Atlantic, where cooler and/or more variable ocean conditions are generally unfavorable for coral growth.

The majority of records in the database fall between 1800 and 2010 CE (Fig. 1b). Approximately 28% of records in the database cover time intervals earlier than the 1800s, with most of these records coming from corals that are dead when



collected (often referred to as "fossil" corals), which provide short, discrete time series often spanning several decades. The oldest such record in the database is a coral from Hainan Island in the South China Sea that covers 167–309 CE (Xiao et al., 2017).

**Figure 2. Total number of records with core top dates between 1900 and 2020 CE, sorted into 5-year bins and organized by (a) ocean basin and (b) available proxy data. Records with core top dates prior to 1900 CE are not shown. East and West Pacific Ocean are split at 180° longitude. (c) Global core top date spatial distribution of coral records in the CoralHydro2k database. Records with core top dates prior to 1975 CE are not shown (31 records).**

A surge in coral-based proxy record generation began in the early 1990s and is reflected in the most common core-top ages occurring in the period from 1990 to 2015 CE (Fig. 2a–b). Peak record density occurs between the late 1980s to early

1990s (Fig. 1b), reflecting increased coral coring efforts in all tropical oceans from 1985–2015 CE (Fig. 2) that increases data

coverage across this interval. Record density precipitously drops after 1998 CE (Fig. 1b), which may simply reflect the 5- to 15-year delay between core collection and record publication. However, we observe that fewer new records are available from more remote regions of the tropics (Fig. 2c). The availability of Sr/Ca and paired records began in the late 1990s (Fig. 2b) with the development of a rapid, high precision, and cost-effective method for measuring Sr/Ca using ICP-OES (Schrag, 1999).

Sr/Ca and paired records in the database that have core top dates prior to the late 1990s typically represent updates or extensions to previously published $\delta^{18}O$ records (e.g., Felis et al., 2000, 2018) or fossil records.

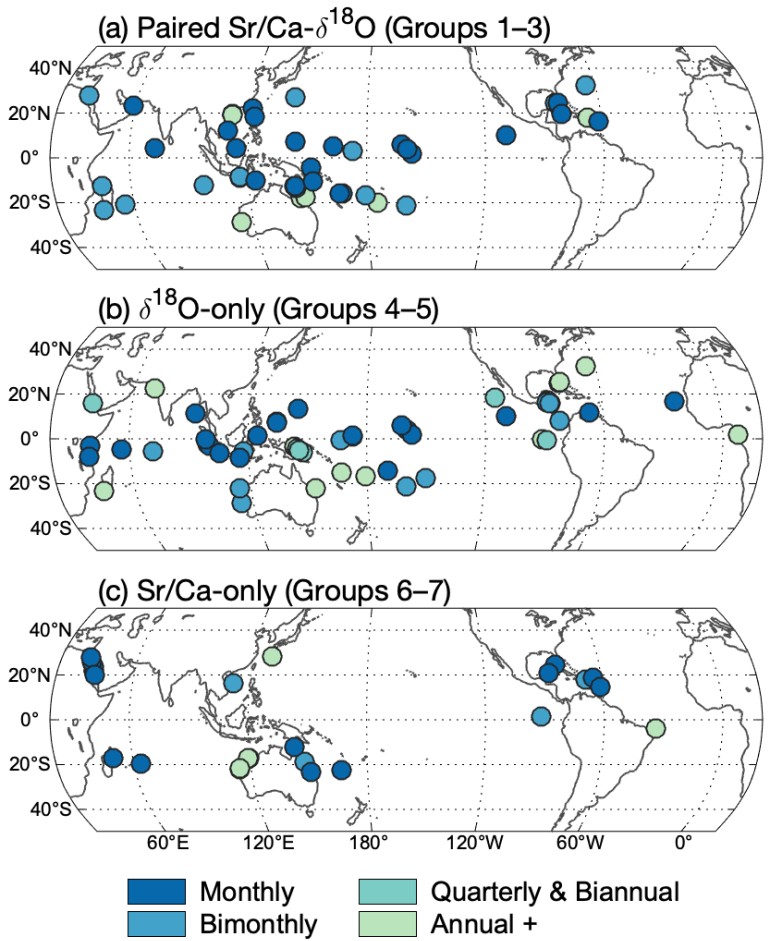

**Figure 3. Resolution of coral records in the CoralHydro2k database. Spatial distributions of temporal resolution for (a) paired Sr/Ca-δ¹⁸O, (b) δ¹⁸O-only, and (c) Sr/Ca-only records.**

A majority of the records included in the CoralHydro2k database offer seasonal or sub-seasonal resolution: 76% of the records in the CoralHydro2k database have monthly or bimonthly resolution, 6% have quarterly to biannual resolution, and 18% have annual or lower resolution (Fig. 3).

Earth System
Science
Data

## 3.2 Relationship to sea surface temperature

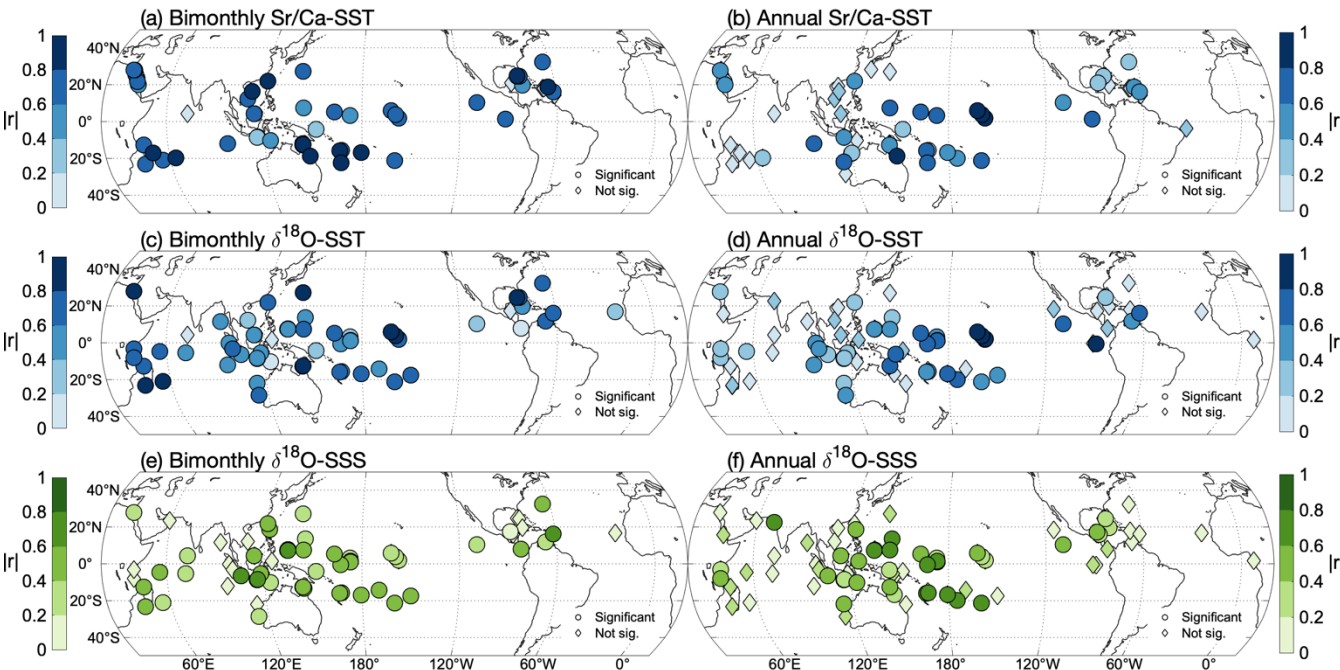

**Figure 4.** Absolute correlations between coral $\delta^{18}O$ and Sr/Ca and local sea surface temperature (SST) from 1950–2020 CE (a-d) and between coral $\delta^{18}O$ and local sea surface salinity (SSS) (e-f) from 1970–2010 CE at bimonthly (left) and annual (April–March; right) resolutions. SST and SSS were taken from the grid box nearest to each coral record in the NOAA ERSSTv5 (Huang et al., 2017) and Hadley EN4 (Good et al., 2013) data sets, respectively. Significant correlations are denoted by circles (greater than 90% confidence interval) and non-significant correlations are denoted by diamonds. We note that significance can vary based on the choice of gridded data set and grid box, annual averaging period and correlation interval, and as such, the values shown here may differ from those reported in the original publications for each record. Correlations are shown as absolute values for ease of visualization, but we note that the linear relationship between SST and coral Sr/Ca or $\delta^{18}O$ is negative.

Proxy records in the CoralHydro2k database capture SST variability on seasonal and longer timescales. To highlight relationships between temperature and proxy records in CoralHydro2k, we calculate Pearson correlation coefficients between the records and local SST (2º grid area) from the NOAA ERSSTv5 data set (Huang et al., 2017). Significance is assessed here at the 90% confidence level: for bimonthly average data, 92% of Sr/Ca and 96% of $\delta^{18}O$ records have a significant correlation with SST. Significant absolute correlations range from 0.23–0.94 (Sr/Ca-SST) and 0.13–0.89 ($\delta^{18}O$-SST) for the interval 1950–2020 CE, with a median correlation of 0.74 for Sr/Ca-SST and 0.59 for $\delta^{18}O$-SST (Fig. 4a,c). Bimonthly average correlations are generally stronger at higher latitudes where the seasonal range in temperature is larger.

Significant absolute correlations between annual-average proxy time series and local SST range from 0.26–0.89 for Sr/Ca data and 0.26–0.92 for $\delta^{18}O$ data, with medians of 0.50 for Sr/Ca-SST and 0.57 for $\delta^{18}O$-SST correlations (Fig. 4b,d). For annual-average data, 43% of Sr/Ca-SST and 56% of $\delta^{18}O$-SST correlations are significant. Here, we use the April–March tropical year for annual averages to avoid splitting large-scale tropical variability between years (Ropelewski and Halpert,

1987). The higher annual proxy-SST correlations occur near the equator, particularly in the central and western tropical Pacific,

where the ENSO drives large SST changes on interannual timescales.

We note that significant discrepancies exist among gridded SST data products (e.g., HadISST, ERSST, OISST) due to the scarcity of observations across space and time and the different statistical techniques used to infill missing data in each SST data product (e.g., Deser et al., 2010; Freeman et al., 2017; Kennedy et al., 2019). Thus, the proxy-SST correlations presented here may deviate from those stated in each record's original publication.

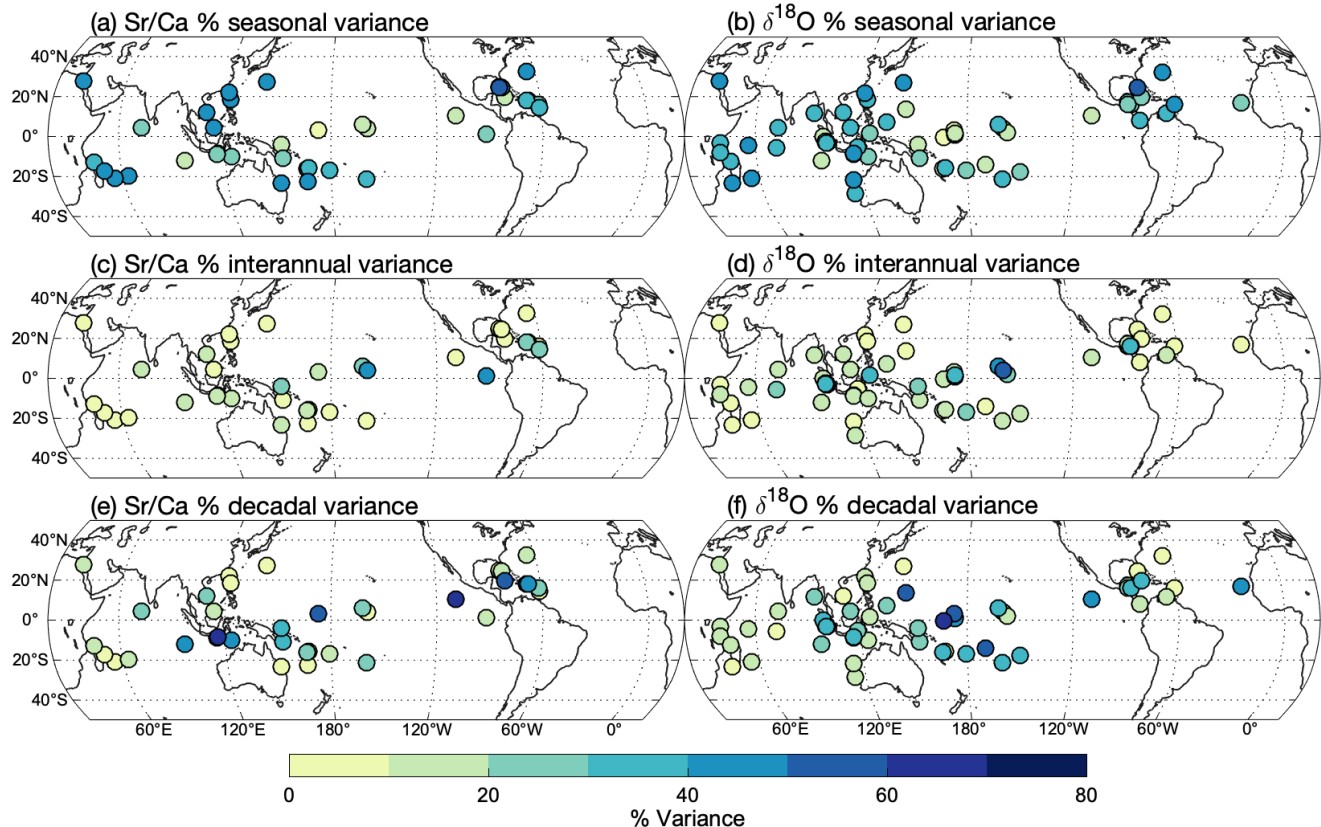

**Figure 5. Percent variance of coral Sr/Ca (a,c,e) and δ¹⁸O (b,d,f) records calculated as the fraction of variance that each time scale of variability contributes to total time series variance. Variance is calculated across the full length of each coral record. (a–b) Highpass variability calculated using a 13-month filter. (c–d) 2–7 year bandpass percent variability that includes interannual variance driven by the El Niño-Southern Oscillation (ENSO). (e–f) 10-year lowpass variability. All percent variability was calculated**

**only for records at least 30 years in length.**

Patterns observed in proxy-SST correlations are also mirrored in the dominant mode of variability displayed by each record. To examine the relative contributions of seasonal, interannual, and decadal variability in coral records, we apply a 13-month highpass (seasonal), a 2–7 year bandpass (interannual), and a 10-year lowpass filter to all monthly and bimonthly records that are at least 30 years long. Filtering was performed using a 6th-order Butterworth filter in MATLAB, with the filter

order used to optimize filtering in the decadal band. Variance for each filtered series is normalized by the proxy record variance

determined for the entire record length to enable comparison between $\delta^{18}O$ and Sr/Ca (Fig. 5). The seasonal variance in both proxies increases with latitude (Fig. 5a-b), with records in the subtropics exhibiting greater seasonal variance than records close to the equator. Conversely, records close to the equator contain higher proportions of interannual variance (Fig. 5c-d). This pattern is more apparent among longer coral $\delta^{18}O$ records, as several Sr/Ca records in the database do not meet the 30-

305  year length requirement for bandpass filtering.

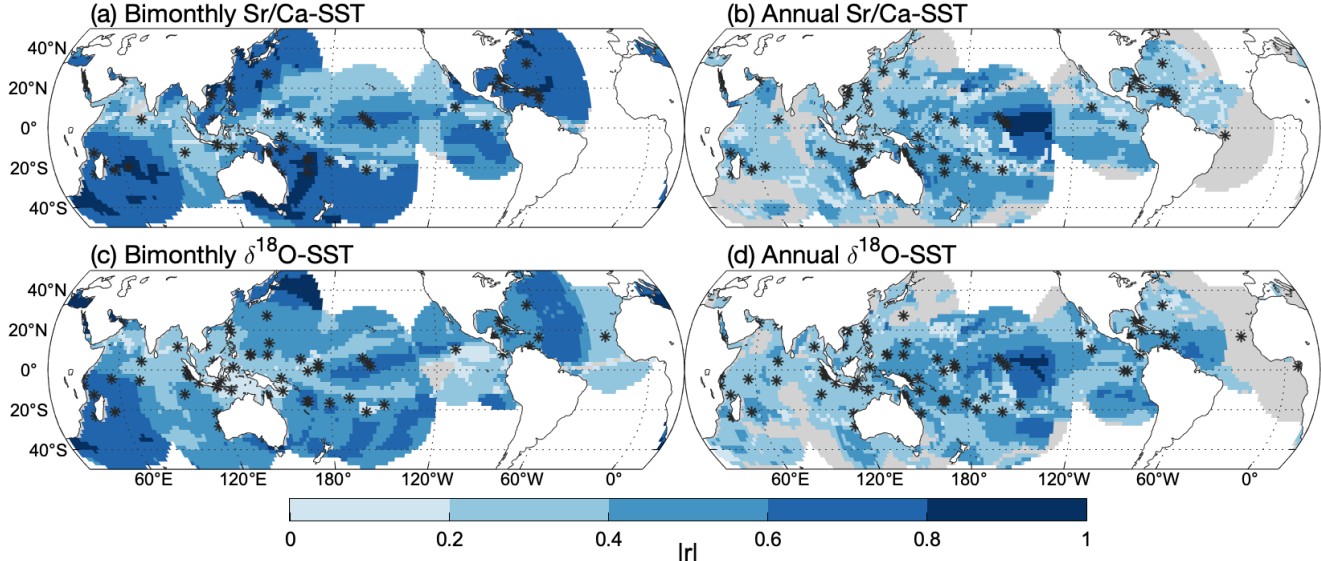

**Figure 6. Median absolute coral proxy-sea surface temperature (SST) correlation within 3000 km of each SST grid box (ERSSTv5). Bimonthly (left) and annual (April–March; right) correlations shown are significant at the 90% confidence level. Grid boxes with records within 3000 km but no significant correlations are shaded gray. Correlations are calculated using available data from 1950–**
**2020 CE. Record locations are indicated by black asterisks.**

For global or regional climate reconstructions, it is useful to consider the relationship of gridded SST data products to the proxy network as opposed to the relationship of individual records to the nearest grid point in those data products. To assess the reconstruction potential of the CoralHydro2k proxy network, we calculate the median absolute correlation between each ERSSTv5 grid box and all available records in the database within a 3,000 km radius (Fig. 6). We find significant

(assuming a 90% confidence interval) annual Sr/Ca-SST correlations across 56% of the tropical and subtropical oceans (Fig. 6b), and significant annual $\delta^{18}O$-SST correlations across 60% (Fig. 6d). Consistent with previous results, bimonthly correlations between SST and both proxies are higher (Fig. 6a,c) due to the seasonal cycle. Whereas non-climatic factors, such as age-model errors (Comboul et al., 2014; Lawman et al., 2020b; Loope et al., 2020), may lower the correlation between coral proxies and SST in some regions, significant correlations observed here highlight the fact the CoralHydro2k database captures

regional to global patterns of climate variability, and thus, is suitable for reconstructing SST variability across much of the tropical and subtropical oceans. Reconstruction potential is limited in the eastern Pacific and eastern Atlantic due to the scarcity of corals from those regions.



### 3.3 Relationship to hydrology

Coral $\delta^{18}O$ records capture combined changes in local SST and $\delta^{18}O_{sw}$, with the latter reflecting the balance among
hydrological processes (e.g., precipitation, evaporation, horizontal and vertical ocean advection). Since observed $\delta^{18}O_{sw}$ data
coverage is limited through space and time (e.g., LeGrande and Schmidt, 2006; Boyer et al., 2018; Breitkreuz et al., 2018), we
compare each coral $\delta^{18}O$ record to SSS from the nearest Hadley EN4.2.1 grid box (Good et al., 2013; Gouretski and Reseghetti,
2010) as both SSS and $\delta^{18}O_{sw}$ variability are driven by similar hydrological processes. However, we do note that the
relationship between these two variables may not be spatiotemporally constant (Conroy et al., 2014, 2017). Significant absolute
correlations between coral $\delta^{18}O$ and SSS between 1970 and 2010 range from 0.16–0.69 at bimonthly resolution and 0.28–0.79
at annual resolution (Fig. 4e-f). The highest correlations occur in the Western Pacific Warm Pool region, where there is stronger
SSS variability due to factors that do not strongly covary with temperature such as terrestrial runoff and ocean mixing (Qu et
al., 2014; Murty et al., 2017, 2018b). For western Pacific sites further away from the Maritime Continent, higher SSS-$\delta^{18}O$
correlations may reflect the strong covariance between SSS and SST, especially on interannual timescales. In contrast, coral
$\delta^{18}O$ records from sites close to the equator in the Indian and central equatorial Pacific oceans exhibit lower $\delta^{18}O$-SSS
correlations, which suggests that SSS variability at these sites is smaller relative to SST or may point to potential biases in
gridded SSS data products.

Many $\delta^{18}O$-SSS correlations at annual resolution are not significant; however, this may be more reflective of the SSS
data set used here rather than the integrity of records in the database. Historical SSS observational records are much shorter
and sparser than SST before the satellite era (Good et al., 2013; Boyer et al., 2018; Friedman et al., 2017), especially in the
tropical and subtropical oceans. Consequently, much larger discrepancies exist among gridded SSS data sets than those found
between gridded SST products (Carton et al., 2018, 2019; Zweng et al., 2019). New and emerging salinity products such as
NASA's Soil Moisture Active Passive (SMAP) Sea Surface Salinity (Vazquez-Cuervo and Gomez-Valdes, 2018), ESA's Soil
Moisture and Ocean Salinity Mission (SMOS; Boutin et al., 2016), Aquarius (Drucker and Riser, 2014), and Argo (Schmid et
al., 2007) will be important calibration data sets for future coral studies or reconstructions that cover the years since 2011 CE.
Nonetheless, the lack of long, historical SSS records highlights the need for independent coral-based constraints on long-term
hydrological trends across the tropical and subtropical oceans.

## 3.4 Local reproducibility of Sr/Ca and δ¹⁸O records

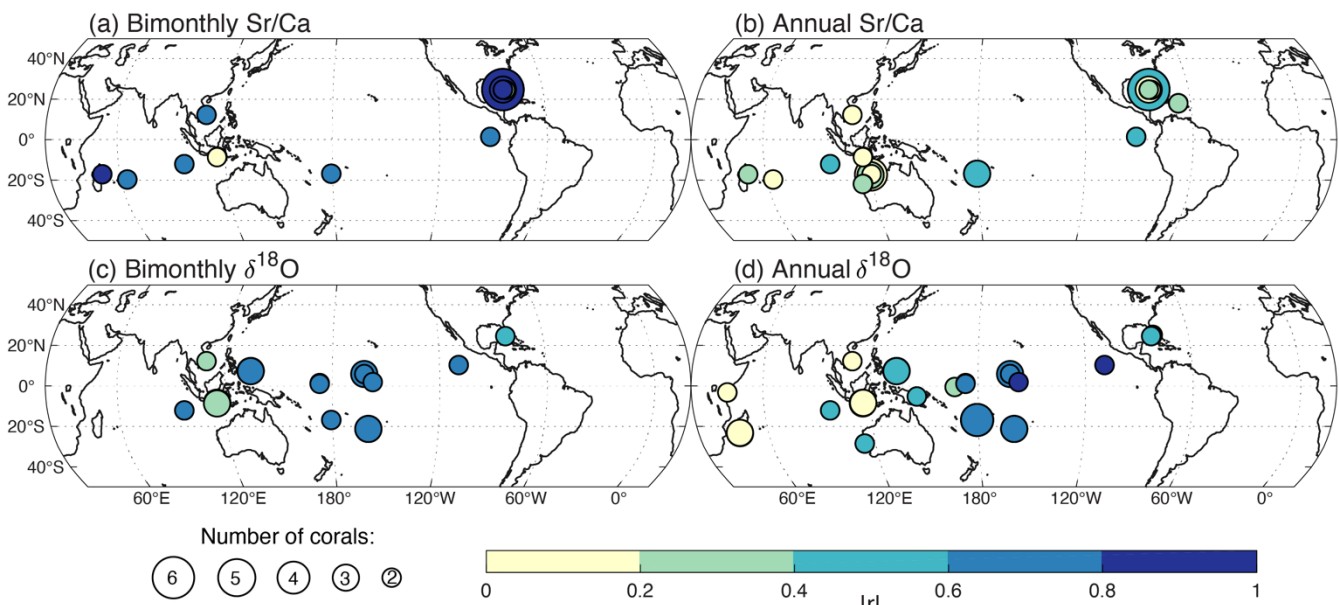

**Figure 7. Median absolute correlation of coral proxy records to other records within a 50 km radius. Correlation was calculated over the common time period between two records, provided that there was a minimum of 20 years of overlap. Marker size indicates the number of records used in the median correlation calculation at each site (largest: N = 6).**

We assess the "local" reproducibility of coral records in the database by comparing each proxy record to records of the same type within a 50 km radius with at least 20 years in common (Fig. 7). As the CoralHydro2k database represents the most comprehensive coral-based proxy compilation effort to date, approximately 36% of the records are within a 50 km radius of one to five contemporaneous records. Bimonthly absolute correlations for Sr/Ca records within 50 km of each other range from 0.11–0.95 (Fig. 7a), and bimonthly δ¹⁸O correlations range from 0.24–0.79 (Fig. 7c). Similarly, annual correlations for Sr/Ca records within 50 km of each other range from 0.04–0.59 (Fig. 7b), and annual correlations for δ¹⁸O records range from 0.01–0.87 (Fig. 7d). Whereas we observe good reproducibility at most sites, the highest degree of local reproducibility among both Sr/Ca and δ¹⁸O records occurs in more open ocean settings (e.g., the central Pacific), where there is less spatial variability in growth environments, ocean advection patterns, local SST, and SSS across short distances.

Reproducibility studies show that proxy records from corals growing on the same reef can exhibit inconsistent mean values (e.g., Giry et al., 2012; Felis et al., 2003, 2004, 2014; DeLong et al., 2007, 2013; Sayani et al., 2021) or proxy-SST relationships (e.g., DeLong et al., 2012; Sayani et al., 2019). The exact causes of intercolony variability are not known. Intercolony differences in mean values are often attributed to subtle differences in reef environments, unresolved interspecies differences, or "vital effects": a catch-all term used to describe a myriad of unknown physiological and/or metabolic processes that impact the incorporation of oxygen isotopes and trace elements into coral skeletons (Weber, 1973; Weber and Woodhead, 1972; McConnaughey, 1989). Proxy-SST relationships may also vary among sites due to different coral micro-sampling



methods (e.g., punch/spot drilling versus continuous micro-milling), the lack of standardized analytical methods for measuring
Sr/Ca prior to the use of the coral reference material JCp-1 (Hathorne et al., 2013), differences in regression methods and
instrumental data sets used to calculate regressions (Corrège, 2006), and other regional parameters (Murty et al., 2018a).
Despite these intercolony differences, contemporaneous proxy records often exhibit similar seasonal and interannual variability
(e.g., Felis et al., 2004; Giry et al., 2012; DeLong et al., 2012, 2014; Kuffner et al., 2017; Sayani et al., 2019), highlighting that
coral records are indeed capturing common climate signals and can be used to reconstruct regional and global climate trends
and variability.

Whereas intercolony variability has mostly been studied in massive *Porites* spp. corals, which are widely distributed
throughout the Indian and Pacific oceans and most commonly used in paleoclimate reconstructions, some species (e.g.,
*Siderastrea siderea,* found in the Atlantic Ocean) exhibit more reproducibility among coral colonies in Sr/Ca, $\delta^{18}$O, and
calibration equations (Maupin et al., 2008; DeLong et al., 2014, 2016; Kuffner et al., 2017; Weerabaddana et al., 2021). More
work is needed to both quantify intercolony variability in different coral species and understand the impact of calibration
method on coral-based temperature reconstructions.

## 4 Usage notes

### 4.1 General applications

The CoralHydro2k database is the most comprehensive compilation of coral $\delta^{18}$O and Sr/Ca records to date. The
database offers extensive coverage of monthly to annually resolved marine proxy records that can be used to investigate near-
surface hydrology and temperature variability across the global tropics and subtropics. Comparable information at similarly
high resolution is rarely available with other marine paleo-archives. Paired coral Sr/Ca-$\delta^{18}$O records allow for independent
reconstruction and investigation of pre-industrial temperature and hydrologic changes at seasonal, interannual, and decadal
time scales. The inclusion of both unpaired and short proxy records, many of which did not meet the selection criteria of
previous PAGES 2k data compilations, allows the CoralHydro2k database to be used for applications beyond large-scale
temperature and hydrology reconstructions. This includes, and is certainly not limited to, proxy calibration studies, proxy-
system model development, and paleo-data assimilation efforts. Records in the CoralHydro2k database can also be compared
to model outputs, either by converting coral Sr/Ca into temperature for direct comparison or by using proxy system modeling
to estimate proxy composition from climate model output. Coral $\delta^{18}$O records and coral-derived $\delta^{18}$O$_{sw}$ records can also be
directly compared with new simulations from isotope-enabled models. A brief overview of how to access, query, and cite the
database is provided in the sections below.

### 4.2 Searching the CoralHydro2k database

Each LiPD serialization of the CoralHydro2k database contains the variables 'D' and 'TS', where 'D' is a site-centric
storage of data and 'TS' is an expanded data structure holding the same information stored in a 'per time series' format.



MATLAB and R serializations also contain an 'sTS' variable to be consistent with other PAGES2k data sets, which contains
the same information as variable 'TS'. Proxy records stored in the CoralHydro2k database can be searched for using a variety
of keywords or parameters. We suggest users initially narrow their search by filtering for groups of interest using the field
*paleoData_coralHydro2kGroup*, which sorts records based on proxy type, record resolution, and temporal coverage. A
detailed summary of group definitions is in Table 1. The database can also be queried by:

• Proxy type, using the field *paleoData_variableName*.

• Temporal coverage, using *minYear* and *maxYear* to search for proxy record start and end years, respectively.

• Record resolution, which can be searched by nominal resolution (Table 3) using *hasResolution_nominal* or numerically
(minimum, mean, median, maximum) using fields beginning with '*hasResolution_*'. See Table 2 for more information.

• Location, using geographic coordinates (*geo_latitude* and *geo_longitude*), site name (*geo_siteName*), or ocean basin
(*geo_ocean*). *geo_ocean* is the level one ocean basin listed in the World Ocean Atlas (Boyer et al., 2018) for the geographic
coordinates of the record.

• Coral species, using *paleoData_archiveSpecies*.

**4.3 Data availability, updates, and versioning**

The development of the CoralHydro2k database was guided by FAIR data principles (Wilkinson et al., 2016), which
strive to make scholarly data Findable, Accessible, Interoperable, and Reusable. Thus, the CoralHydro2k database employs
the LiPD framework (McKay and Emile-Geay, 2016), a standardized, machine-readable format for archiving and describing
paleoclimate data, with serializations for MATLAB, R, and Python available at https://doi.org/10.25921/yp94-v135 (Walter
et al., 2022). Also available on the database website is a MATLAB example script to help new users search the database.

One of CoralHydro2k's core goals is to create an actively curated coral database. We encourage the community to
submit newly published coral $\delta^{18}O$ and Sr/Ca records using the data submission form located on the repository website linked
above. Newly published records will be compiled and added to the database on an annual basis. Updates to the database will
follow the versioning scheme used by the PAGES2k database (PAGES2k Consortium, 2017). The first release of the
CoralHydro2k database is version 1.0.0. The version number has three counters in the following form $C_1.C_2.C_3$. The first
counter, $C_1$, is updated with each publication of a formal update of the data set.  The second counter, $C_2$, is updated when a
record is added or removed. The third counter, $C_3$, is updated when a modification is made to the data or metadata. It is
anticipated that future versions and a change log describing updates with each new version will be made available at the same
location as the original data release.

**4.4 Citation**

Researchers utilizing the whole CoralHydro2k database or a significant portion of the database should cite this paper
and the paper describing the most recent version of the database. If only a small subset of the records is being used, researchers



should also cite the papers that originally describe each coral record used. Citation information associated with each record in the database as well as a link to the original public archive of each data set is included in the metadata to facilitate users in crediting the original data generators in their use of the coral data.

## 5 Conclusion

Shallow-water corals provide monthly to annually resolved climate records from data-scarce locations across the tropical and subtropical oceans and are incredibly useful for extending modern-day observations back into the preindustrial era, contextualizing anthropogenic climate trends, and improving the skill of future climate projections. The PAGES CoralHydro2k project was formed to facilitate the use of coral paleoclimate records by the broader scientific community. Our first effort on this front is the CoralHydro2k database: a mostly unfunded endeavor representing the collective efforts of 40+

researchers across different career stages, institutes, and time zones, meeting monthly to bi-weekly and working asynchronously over the past four years. Subsequent publications from the CoralHydro2k project will use this database to evaluate proxy-SST calibrations and methodological differences used in coral-based climate reconstructions as well as investigate past tropical ocean hydroclimate trends using data assimilation and comparison to isotope enabled-models. Furthermore, the CoralHydro2k team has also been collecting instrumental seawater $\delta^{18}$O data as part of our database

compilation efforts. That database will be released in the near future — also following the FAIR standards — and will also be maintained with active curation (see DeLong et al., in press). While the fruits of the CoralHydro2k database are likely to come over the next 5–10 years, continuing to invest as a community in compiling standardized data sets will inevitably elevate the utility of each record.

The CoralHydro2k database is a comprehensive, machine-readable, standardized, and actively curated database of

coral $\delta^{18}$O and Sr/Ca records. Records in the CoralHydro2k database track large-scale regional SST and hydrology signals across seasonal, interannual, and decadal timescales with a high degree of reproducibility. As such, the records in the database can be used for investigating tropical and subtropical SST and hydrology variability on societally relevant time scales and can be combined with large networks of terrestrial paleo-archives of climate variability such as tree rings, ice cores, or speleothems to investigate past and present ocean-atmosphere-land interactions. Moreover, the database enables global-scale comparisons

of coral-based paleoclimate reconstructions with state-of-the-art climate models, either through the use of forward models (Thompson et al., 2011; Dee et al., 2015, 2017; Tardif et al., 2019), or directly in the case of isotope-enabled models (Konecky et al., 2020). The comprehensive and high-resolution nature of the CoralHydro2k database also makes it ideally suited as an input database for paleoclimate data assimilation efforts such as the Last Millennium Reanalysis (Hakim et al., 2016; Steiger et al., 2018; Tardif et al., 2019; Sanchez et al., 2021).




**Appendix A**

**Table A1. Reference table of publications sited in the CoralHydro2k database. Citations in the *Cited publications* column are listed in the order presented in the database (*pub1*, *pub2*, *pub3*).**

| Unique ID | Group | Proxies | Cited publications | Latitude | Longitude | Location |
|---|---|---|---|---|---|---|
| *Atlantic Ocean* | | | | | | |
| AL16PUR01 | 6 | Sr/Ca | (Alpert et al., 2017) | 18.1153 | -67.9374 | Mona Island, Puerto Rico |
| AL16PUR02 | 6 | Sr/Ca | (Alpert et al., 2017) | 17.93 | -67.01 | Pinacles Reef, Puerto Rico |
| AL16YUC01 | 6 | Sr/Ca | (Alpert et al., 2017) | 20.8321 | -86.8789 | Puerto Morelos, Mexico (Yucatan Peninsula) |
| CA13DIA01 | 4 | d18O | (Carilli et al., 2013) | 16.064 | -86.951 | Diamond Caye, Utila, Honduras (Gulf of Honduras) |
| CA13PEL01 | 4 | d18O | (Carilli et al., 2013) | 15.978 | -86.485 | Cayos Cochinos, Honduras (Gulf of Honduras) |
| CA13SAP01 | 4 | d18O | (Carilli et al., 2013) | 16.129 | -88.25 | Sapodilla Cayes, Belize (Gulf of Honduras) |
| CA13TUR01 | 4 | d18O | (Carilli et al., 2013) | 17.307 | -87.801 | Turneffe Atoll, Belize (Gulf of Honduras) |
| DE14DTO01 | 6 | Sr/Ca | (DeLong et al., 2014, 2016; Flannery et al., 2017) | 24.6988 | -82.7974 | Dry Tortugas, Florida, USA (Pulaski Reef, Florida Keys) |
| DE14DTO02 | 6 | Sr/Ca | (DeLong et al., 2014, 2016; Flannery et al., 2017) | 24.617 | -82.867 | Dry Tortugas, Florida, USA (south of Long Key, Florida Keys) |
| DE14DTO03 | 6 | Sr/Ca | (DeLong et al., 2014, 2016, 2011) | 24.6988 | -82.7974 | Dry Tortugas, Florida, USA (Pulaski Reef, Florida Keys) |
| DE14DTO04 | 6 | Sr/Ca | (DeLong et al., 2014, 2016, 2011) | 24.6949 | -82.7947 | Dry Tortugas, Florida, USA (Pulaski Reef, Florida Keys) |
| DR00KSB01 | 5 | d18O | (Draschba et al., 2000) | 32.467 | -64.568 | Kitchen Shoals, Bermuda (Sargasso Sea) |



| DR00NBB01 | 5 | d18O | (Draschba et al., 2000) | 32.5 | -64.7 | Northeast Breakers, Bermuda (Sargasso Sea) |
| EV18ROC01 | 7 | Sr/Ca | (Evangelista et al., 2018) | -3.86 | -33.77 | Rocas Atoll, Rio Grande do Norte, Brazil |
| FL17DTO01 | 6 | Sr/Ca | (Flannery et al., 2017; Flannery and Poore, 2013; DeLong et al., 2011) | 24.6986 | -82.7986 | Dry Tortugas, Florida, USA (Pulaski Reef, Florida Keys) |
| FL17DTO02 | 6 | Sr/Ca | (Flannery et al., 2017; Weinzierl et al., 2016) | 24.699 | -82.799 | Dry Tortugas, Florida, USA (Pulaski Reef, Florida Keys) |
| FL18DTO01 | 6 | Sr/Ca | (Flannery et al., 2018, 2017; Flannery and Poore, 2013) | 24.6949 | -82.7983 | Dry Tortugas, Florida, USA (Pulaski Reef, Florida Keys) |
| FL18DTO02 | 6 | Sr/Ca | (Flannery et al., 2018; Hickey et al., 2013) | 24.6946 | -82.7949 | Dry Tortugas, Florida, USA (Pulaski Reef, Florida Keys) |
| FL18DTO03 | 6 | Sr/Ca | (Flannery et al., 2018; Weinzierl et al., 2016) | 24.703 | -82.848 | Dry Tortugas, Florida, USA (near North Key Harbor, Florida Keys) |
| FL18DTO04 | 6 | Sr/Ca | (Flannery et al., 2018; Weinzierl et al., 2016) | 24.703 | -82.844 | Dry Tortugas, Florida, USA (near North Key Harbor, Florida Keys) |
| GO08BER01 | 1 | d18O, Sr/Ca | (Goodkin et al., 2008, 2005; Goodkin, 2007) | 32.33 | -64.68 | Bermuda |
| HE08LRA01 | 4 | d18O | (Hetzinger et al., 2008) | 11.77 | -66.75 | Cayo Sal, Los Roques Archipelago, Venezuela |
| HE10GUA01 | 1 | d18O, Sr/Ca | (Hetzinger et al., 2010, 2006) | 16.2 | -61.49 | Isle de Gosier, Guadeloupe (Lesser Antilles) |
| KI08PAR01 | 3 | d18O, Sr/Ca | (Kilbourne et al., 2008, 2010) | 17.93 | -67 | La Parguera, Puerto Rico (Turrumote Reef) |

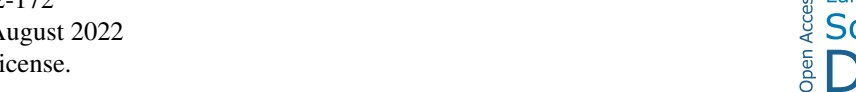

| | | | | | | |
|---|---|---|---|---|---|---|
| KI14PAR01 | 3 | d18O, Sr/Ca | (Kilbourne et al., 2014; Watanabe et al., 2001; Kilbourne et al., 2008) | 18 | -67 | La Parguera, Puerto Rico (Turrumote Reef) |
| MA08DTO01 | 2 | d18O, Sr/Ca | (Maupin et al., 2008; DeLong et al., 2016) | 24.6167 | -82.8667 | Dry Tortugas, Florida, USA (Long Key, Dry Tortugas) |
| MO06PED01 | 4 | d18O | (Moses et al., 2006) | 16.76 | -22.89 | Pedra de Lume, Sal Island (Cape Verde Islands) |
| RE18CAY01 | 1 | d18O, Sr/Ca | (von Reumont et al., 2018, 2016) | 19.7 | -80.06 | Little Cayman, Cayman Islands |
| RO19MAR01 | 6 | Sr/Ca | (Rodriguez et al., 2019) | 14.4512 | -60.929 | Grande Cai, Martinique |
| RO19PAR01 | 7 | Sr/Ca | (Rodriguez et al., 2019; Kilbourne et al., 2010; Watanabe et al., 2001) | 17.9368 | -67.0184 | Parguera, Puerto Rico |
| RO19YUC01 | 7 | Sr/Ca | (Rodriguez et al., 2019; Vásquez-Bedoya et al., 2012) | 20.8321 | -86.8789 | Puerto Morelos, Mexico (Yucatan Peninsula) |
| SM06LKF01 | 2 | d18O, Sr/Ca | (Smith et al., 2006) | 24.56 | -81.41 | Looe Key, Florida, USA (Florida Keys) |
| SM06LKF02 | 2 | d18O, Sr/Ca | (Smith et al., 2006) | 24.56 | -81.41 | Looe Key, Florida, USA (Florida Keys) |
| SW98STP01 | 5 | d18O | (Swart et al., 1998) | 1.67 | 7.58 | Ponta Banana, Principe Island (Gulf of Guinea) |
| SW99LIG01 | 5 | d18O | (Swart et al., 1999) | 25.23 | -80.4167 | Lignumvitae Basin, Florida, USA (Florida Bay) |
| SW99LIG02 | 5 | d18O | (Swart et al., 1999, 1996) | 25 | -80.6 | Lignumvitae Basin, Florida Bay (Florida Bay) |
| XU15BVI01 | 6 | Sr/Ca | (Xu et al., 2015) | 18.72 | -64.3167 | Anegada, British Virgin Islands (Soldier Point) |
| XU15BVI02 | 6 | Sr/Ca | (Xu et al., 2015) | 18.72 | -64.3167 | Anegada, British Virgin Islands (Soldier Point) |



| XU15BVI03 | 6 | Sr/Ca | (Xu et al., 2015) | 18.72 | -64.3167 | Anegada, British Virgin Islands (Soldier Point) |
|---|---|---|---|---|---|---|
| ***Pacific Ocean*** | | | | | | |
| AS05GUA01 | 4 | d18O | (Asami et al., 2005) | 13.598 | 144.836 | Double Reef, Guam |
| BA04FIJ01 | 3 | d18O, Sr/Ca | (Bagnato et al., 2004) | -16.82 | 179.23 | Savusavu Bay, Vanua Levu, Fiji |
| BA04FIJ02 | 2 | d18O, Sr/Ca | (Bagnato et al., 2004) | -16.82 | 179.23 | Savusavu Bay, Vanua Levu, Fiji |
| BO14HTI01 | 2 | d18O, Sr/Ca | (Bolton et al., 2014; Goodkin et al., 2021) | 12.21 | 109.31 | Hon Tre Island, Vietnam |
| BO14HTI02 | 2 | d18O, Sr/Ca | (Bolton et al., 2014; Goodkin et al., 2021) | 12.21 | 109.31 | Hon Tre Island, Vietnam |
| BO99MOO01 | 4 | d18O | (Boiseau et al., 1999, 1998) | -17.5 | -149.83 | Moorea, French Polynesia |
| CA07FLI01 | 3 | d18O, Sr/Ca | (Calvo et al., 2007) | -17.73 | 148.43 | Flinders Reef, Australia (Coral Sea) |
| CA14BUT01 | 2 | d18O, Sr/Ca | (Carilli et al., 2014) | 3.2 | 172.8 | Butaritari Atoll, Republic of Kiribati (Gilbert Islands) |
| CH03BUN01 | 4 | d18O | (Charles et al., 2003) | 1.5 | 124.83 | Bunaken Island, Indonesia (North Sulawesi) |
| CH03LOM01 | 4 | d18O | (Charles et al., 2003) | -8.25 | 115.5 | Padang Bai, Bali, Indonesia (Lombok Strait) |
| CH18YOA01 | 6 | Sr/Ca | (Chen et al., 2018) | 16.448 | 111.605 | Lingyang Reef, Yongle Atoll (South China Sea) |
| CH18YOA02 | 6 | Sr/Ca | (Chen et al., 2018) | 16.448 | 111.605 | Lingyang Reef, Yongle Atoll (South China Sea) |
| CO03PAL01 | 4 | d18O | (Cobb et al., 2003a, b) | 5.87 | -162.13 | Palmyra Island, United States Minor Outlying Islands (Line Islands) |
| CO03PAL02 | 4 | d18O | (Cobb et al., 2003a, b) | 5.87 | -162.13 | Palmyra Island, United States Minor Outlying Islands (Line Islands) |





| | | | | | | |
|---|---|---|---|---|---|---|
| CO03PAL03 | 4 | d18O | (Cobb et al., 2003a, b) | 5.87 | -162.13 | Palmyra Island, United States Minor Outlying Islands (Line Islands) |
| CO03PAL04 | 4 | d18O | (Cobb et al., 2003a, b) | 5.87 | -162.13 | Palmyra Island, United States Minor Outlying Islands (Line Islands) |
| CO03PAL05 | 4 | d18O | (Cobb et al., 2003a, b) | 5.87 | -162.13 | Palmyra Island, United States Minor Outlying Islands (Line Islands) |
| CO03PAL06 | 4 | d18O | (Cobb et al., 2003a, b) | 5.87 | -162.13 | Palmyra Island, United States Minor Outlying Islands (Line Islands) |
| CO03PAL07 | 4 | d18O | (Cobb et al., 2003a, b) | 5.87 | -162.13 | Palmyra Island, United States Minor Outlying Islands (Line Islands) |
| CO03PAL08 | 4 | d18O | (Cobb et al., 2003a, b) | 5.87 | -162.13 | Palmyra Island, United States Minor Outlying Islands (Line Islands) |
| CO03PAL09 | 4 | d18O | (Cobb et al., 2003a, b) | 5.87 | -162.13 | Palmyra Island, United States Minor Outlying Islands (Line Islands) |
| CO03PAL10 | 4 | d18O | (Cobb et al., 2003a, b) | 5.87 | -162.13 | Palmyra Island, United States Minor Outlying Islands (Line Islands) |
| CO93TAR01 | 4 | d18O | (Cole et al., 1993) | 1.42 | 173.03 | Tarawa Atoll, Republic of Kiribati (Gilbert Islands) |
| DE12ANC01 | 6 | Sr/Ca | (DeLong et al., 2012, 2007) | -22.48 | 166.46 | Amedee Island, New Caledonia |
| DE13HAI01 | 3 | d18O, Sr/Ca | (Deng et al., 2013) | 19.29 | 110.656 | Longwan, Qionghai, China (Hainan Island) |
| DO18DAV01 | 6 | Sr/Ca | (D'Olivo et al., 2018) | -18.8 | 147.63 | Davies Reef, Australia (Great Barrier Reef) |



| DR99ABR01 | 5 | d18O | (Druffel and Griffin, 1999, 1993) | -22.1 | 153 | Abraham Reef, Australia (Great Barrier Reef) |
|---|---|---|---|---|---|---|
| DU94URV01 | 5 | d18O | (Dunbar et al., 1994) | -0.4084 | -91.234 | Urvina Bay, Isabela Island, Ecuador (Galapagos Islands) |
| DU94URV02 | 5 | d18O | (Dunbar et al., 1994) | -0.4084 | -91.234 | Urvina Bay, Isabela Island, Ecuador (Galapagos Islands) |
| EV98KIR01 | 4 | d18O | (Evans et al., 1998) | 2 | -157.3 | Kiritimati (Christmas) Island, Republic of Kiribati (Line Islands) |
| FE09OGA01 | 1 | d18O, Sr/Ca | (Felis et al., 2009) | 27.1059 | 142.1941 | Ogasawara Islands, Japan (Chichijima) |
| GO12SBV01 | 1 | d18O, Sr/Ca | (Gorman et al., 2012; Lawman et al., 2020a) | -15.94 | 166.07 | Sabine Bank, Vanuatu |
| GU99NAU01 | 5 | d18O | (Guilderson and Schrag, 1999) | -0.54 | 166.97 | Nauru Island, Republic of Nauru |
| GU99NAU02 | 4 | d18O | (Guilderson and Schrag, 1999) | -0.54 | 166.97 | Nauru Island, Republic of Nauru |
| HE02GBR01 | 3 | d18O, Sr/Ca | (Hendy et al., 2002) | -17.78 | 146.13 | Central Great Barrier Reef, Australia (Great Barrier Reef) |
| HE13MIS01 | 2 | d18O, Sr/Ca | (Hereid et al., 2013) | -10.69 | 152.81 | Misima Island, Papua New Guinea |
| HE13MIS02 | 2 | d18O, Sr/Ca | (Hereid et al., 2013) | -10.69 | 152.81 | Misima Island, Papua New Guinea |
| JI18GAL01 | 6 | Sr/Ca | (Jimenez et al., 2018) | 1.386 | -91.832 | Shark Bay, Wolf Island, Ecuador (Galapagos Islands) |
| JI18GAL02 | 6 | Sr/Ca | (Jimenez et al., 2018) | 1.386 | -91.832 | Shark Bay, Wolf Island, Ecuador (Galapagos Islands) |





| KA17RYU01 | 7 | Sr/Ca | (Kawakubo et al., 2017, 2014) | 28.3 | 130 | Kikai Island, Japan (Ryukyu Islands) |
|---|---|---|---|---|---|---|
| KI04MCV01 | 2 | d18O, Sr/Ca | (Kilbourne et al., 2004b, a) | -15.7 | 167.2 | Espiritu Santo Island, Vanuatu (Malo Channel) |
| KR20SAR01 | 2 | d18O, Sr/Ca | (Krawczyk et al., 2020) | 4.2922 | 113.8259 | Sarawak, Malaysia (Miri-Sibuti Coral Reefs National Park) |
| KR20SAR02 | 2 | d18O, Sr/Ca | (Krawczyk et al., 2020) | 4.3433 | 113.8983 | Sarawak, Malaysia (Miri-Sibuti Coral Reefs National Park) |
| LI00RAR01 | 1 | d18O, Sr/Ca | (Linsley et al., 2000; Ren et al., 2003; Linsley et al., 2004) | -21.24 | -159.83 | Rarotonga, Cook Islands |
| LI04FIJ01 | 1 | d18O, Sr/Ca | (Linsley et al., 2004) | -16.82 | 179.23 | Vanua Levu, Fiji (Savusavu Bay) |
| LI06FIJ01 | 5 | d18O | (Linsley et al., 2006) | -16.82 | 179.23 | Savusavu Bay, Vanua Levu, Fiji |
| LI06RAR01 | 4 | d18O | (Linsley et al., 2004, 2006) | -21.2378 | -159.828 | Rarotonga, Cook Islands |
| LI06RAR02 | 4 | d18O | (Linsley et al., 2004, 2006) | -21.2378 | -159.828 | Rarotonga, Cook Islands |
| LI94SEC01 | 4 | d18O | (Linsley et al., 1994) | 7.983 | -82.05 | Secas Island, Panama (Gulf of Chiriqui) |
| LI99CLI01 | 4 | d18O | (Linsley et al., 1999) | 10.3 | -109.22 | Clipperton Island |
| MC04PNG01 | 5 | d18O | (McGregor and Gagan, 2004; McGregor et al., 2008) | -3.4118 | 143.637 | Muschu Island, Papua New Guinea |
| MC11KIR01 | 4 | d18O | (McGregor et al., 2011) | 2 | -157.3 | Kiritimati (Christmas) Island, Republic of Kiribati (Line Islands) |
| MO20KOI01 | 2 | d18O, Sr/Ca | (Mohtar et al., 2021) | 5.3 | 163 | Kosrae Island, Fed. States of Micronesia |





| MO20WOA01 | 2 | d18O, Sr/Ca | (Mohtar et al., 2021) | 7.4 | 144 | Wolei Atoll, Fed. States of Micronesia |
| MU17DOA01 | 4 | d18O | (Murty et al., 2017) | -5.382 | 117.914 | Doangdoangan Besar, Indonesia (Makassar Strait) |
| MU18GSI01 | 1 | d18O, Sr/Ca | (Murty et al., 2018b) | -8.38 | 115.71 | Gili Selang, Bali, Indonesia (Lombok Strait) |
| NU09FAN01 | 2 | d18O, Sr/Ca | (Nurhati et al., 2009) | 3.85 | -159.35 | Tabuaeran (Fanning Island), Republic of Kiribati (Line Islands) |
| NU09KIR01 | 2 | d18O, Sr/Ca | (Nurhati et al., 2009) | 1.8667 | -157.4 | Kiritimati (Christmas) Island, Republic of Kiribati (Line Islands) |
| NU11PAL01 | 1 | d18O, Sr/Ca | (Nurhati et al., 2011, 2009; Cobb et al., 2001) | 5.867 | -162.133 | Palmyra Island, United States Minor Outlying Islands (Line Islands) |
| OS13NGP01 | 4 | d18O | (Osborne et al., 2013) | 7.4064 | 134.4353 | Ngaragabel, Palau |
| OS13NLP01 | 4 | d18O | (Osborne et al., 2013) | 7.6569 | 134.5651 | Ngeralang, Palau |
| OS14RIP01 | 4 | d18O | (Osborne et al., 2014, 2013) | 7.2708 | 134.3837 | Rock Islands, Palau |
| OS14UCP01 | 4 | d18O | (Osborne et al., 2014, 2013) | 7.2859 | 134.2503 | Ulong Channel, Palau |
| QU06RAB01 | 1 | d18O, Sr/Ca | (Quinn et al., 2006) | -4.18 | 151.98 | Rabaul, East New Britain, Papua New Guinea |
| QU96ESV01 | 5 | d18O | (Quinn et al., 1996, 1993) | -15 | 167 | Espiritu Santo Island, Vanuatu |
| RA19PAI01 | 1 | d18O, Sr/Ca | (Ramos et al., 2019) | 18.54 | 122.15 | Palaui Island, Philippines (Luzon Strait) |
| RA20TAI01 | 1 | d18O, Sr/Ca | (Ramos et al., 2020) | 21.9 | 120.7 | Houbihu, Taiwan (Luzon Strait) |
| RE19GBR01 | 2 | d18O, Sr/Ca | (Reed et al., 2019) | -12.5 | 143.52 | Eel Reef, Australia (Great Barrier Reef) |
| RE19GBR02 | 2 | d18O, Sr/Ca | (Reed et al., 2019) | -12.6 | 143.3 | Portland Roads, Australia (Great Barrier Reef) |





| | | | | | | |
|---|---|---|---|---|---|---|
| RE19GBR03 | 2 | d18O, Sr/Ca | (Reed et al., 2019) | -13.33 | 143.95 | Reef 13-050, Australia (Great Barrier Reef) |
| RE19GBR04 | 6 | Sr/Ca | (Reed et al., 2019) | -12.09 | 143.29 | Nomad Reef, Australia (Great Barrier Reef) |
| RE19GBR05 | 6 | Sr/Ca | (Reed et al., 2019) | -11.97 | 143.28 | Clerke Reef, Australia (Great Barrier Reef) |
| SA16CLA01 | 5 | d18O | (Sanchez et al., 2016) | 18.4 | -114.7 | Clarion Island, Mexico (Revillagigedos Archipelago) |
| SA18GBR01 | 6 | Sr/Ca | (Saha et al., 2018, 2021) | -23.15 | 150.97 | Great Keppel Island, Australia (Great Barrier Reef) |
| SA19PAL01 | 2 | d18O, Sr/Ca | (Sayani et al., 2019) | 5.878 | -162.142 | Palmyra Island, United States Minor Outlying Islands (Line Islands) |
| SA19PAL02 | 2 | d18O, Sr/Ca | (Sayani et al., 2019) | 5.878 | -162.142 | Palmyra Island, United States Minor Outlying Islands (Line Islands) |
| SA20FAN01 | 4 | d18O | (Sanchez et al., 2020) | 3.85 | -159.35 | Tabuaeran (Fanning Island), Republic of Kiribati (Line Islands) |
| SA20FAN02 | 4 | d18O | (Sanchez et al., 2020) | 3.85 | -159.35 | Tabuaeran (Fanning Island), Republic of Kiribati (Line Islands) |
| SH92PUN01 | 5 | d18O | (Shen et al., 1992) | -0.67 | -89.17 | Punta Pitt, Isla San Cristobal, Ecuador (Galapagos Islands) |
| TA18TAS01 | 4 | d18O | (Tangri et al., 2018) | -14.27 | -169.5 | Ta'u, American Samoa |
| TU01DEP01 | 5 | d18O | (Tudhope et al., 2001) | -5.217 | 145.817 | Madang Lagoon, Papua New Guinea (Deplik Tabat Reef) |
| TU01LAI01 | 5 | d18O | (Tudhope et al., 2001) | -4.15 | 144.883 | Laing Island, Papua New Guinea |





| | | | | | | |
|---|---|---|---|---|---|---|
| TU01SIA01 | 5 | d18O | (Tudhope et al., 2001) | -6.08 | 147.6 | Sialum, Huon Peninsula, Papua New Guinea |
| TU95MAD01 | 5 | d18O | (Tudhope et al., 1995) | -5.22 | 145.82 | Madang Lagoon, Papua New Guinea |
| UR00MAI01 | 4 | d18O | (Urban et al., 2000) | 1 | 173 | Maiana, Republic of Kiribati (Gilbert Islands) |
| WE09ARR01 | 3 | d18O, Sr/Ca | (Wei et al., 2009) | -16.72 | 146.03 | Arlington Reef, Australia (Great Barrier Reef) |
| WU13TON01 | 3 | d18O, Sr/Ca | (Wu et al., 2013; Linsley et al., 2008) | -19.9333 | -174.717 | Ha'afera, Tonga |
| WU14CLI01 | 1 | d18O, Sr/Ca | (Wu et al., 2014; Linsley et al., 1999) | 10.3 | -109.22 | Clipperton Island |
| XI17HAI01 | 3 | d18O, Sr/Ca | (Xiao et al., 2017) | 19.395 | 110.753 | Fengjiawan, Wenchang, China (Hainan Island) |

*Indian Ocean and Bay of Bengal*

| | | | | | | |
|---|---|---|---|---|---|---|
| AB08MEN01 | 4 | d18O | (Abram et al., 2008) | -0.13 | 98.52 | Mentawai Islands, Indonesia (West Sumatra) |
| AB15BHB01 | 4 | d18O | (Abram et al., 2015) | -6.53 | 105.63 | Batu Hitam Beach, Indonesia (Sunda Strait) |
| AB20MEN01 | 4 | d18O | (Abram et al., 2020, 2015) | -3.18 | 100.517 | Mentawai Islands, Indonesia (Tinopo) |
| AB20MEN02 | 4 | d18O | (Abram et al., 2020; Gagan et al., 2015) | -2.37 | 99.745 | Mentawai Islands, Indonesia (Siruamata) |
| AB20MEN03 | 4 | d18O | (Abram et al., 2020) | -3.126 | 100.309 | Mentawai Islands, Indonesia (Saomang) |
| AB20MEN04 | 4 | d18O | (Abram et al., 2020) | -2.752 | 99.995 | Mentawai Islands, Indonesia (Silabu) |
| AB20MEN05 | 4 | d18O | (Abram et al., 2020) | -3.037 | 100.231 | Mentawai Islands, Indonesia (Pororogat) |
| AB20MEN06 | 4 | d18O | (Abram et al., 2020) | -3.0366 | 100.2307 | Mentawai Islands, Indonesia (Pororogat) |
| AB20MEN07 | 4 | d18O | (Abram et al., 2020) | -3.1261 | 100.3097 | Mentawai Islands, Indonesia (Saomang) |



| | | | | | | |
|---|---|---|---|---|---|---|
| AB20MEN08 | 4 | d18O | (Abram et al., 2020) | -3.1261 | 100.3098 | Mentawai Islands, Indonesia (Saomang) |
| AB20MEN09 | 4 | d18O | (Abram et al., 2020) | -3.1259 | 100.3094 | Mentawai Islands, Indonesia (Saomang) |
| CA14TIM01 | 1 | d18O, Sr/Ca | (Cahyarini et al., 2014) | -10.2 | 123.51 | Timor, Indonesia (Ombai Strait) |
| CH97BVB01 | 4 | d18O | (Charles et al., 1997) | -4.6162 | 55.817 | Mahe Island, Republic of the Seychelles (Beau Vallon Bay) |
| CH98PIR01 | 5 | d18O | (Chakraborty and Ramesh, 1998) | 22.6 | 70 | Pirotan Island, Gujarat, India (Gulf of Kutch, Northern Arabian Sea) |
| CO00MAL01 | 5 | d18O | (Cole et al., 2000; Fleitmann et al., 2007) | -3.26 | 40.14 | Malindi Marine Park, Kenya |
| DA06MAF01 | 4 | d18O | (Damassa et al., 2006) | -8.0167 | 39.5 | Fungu Mrima Reef, Tanzania (Mafia Archipelago, Bwejuu Island) |
| DA06MAF02 | 4 | d18O | (Damassa et al., 2006) | -8.0167 | 39.5 | Fungu Mrima Reef, Tanzania (Mafia Archipelago, Bwejuu Island) |
| GR13MAD01 | 6 | Sr/Ca | (Grove et al., 2013) | -17.095 | 49.858 | Nosy Boraha, Madagascar (formerly Ile Sainte-Marie) |
| GR13MAD02 | 6 | Sr/Ca | (Grove et al., 2013) | -17.089 | 49.861 | Nosy Boraha, Madagascar (formerly Ile Sainte-Marie) |
| HE18COC01 | 1 | d18O, Sr/Ca | (Hennekam et al., 2018) | -12.0875 | 96.8752 | Cocos (Keeling) Islands, Australia |
| HE18COC02 | 2 | d18O, Sr/Ca | (Hennekam et al., 2018) | -12.095 | 96.8805 | Cocos (Keeling) Islands, Australia |
| KU00NIN01 | 4 | d18O | (Kuhnert et al., 2000) | -21.905 | 113.965 | Ningaloo Reef, Australia (Western Australia) |
| KU99HOU01 | 4 | d18O | (Kuhnert et al., 1999; Zinke et al., 2014a) | -28.4617 | 113.7683 | Houtman Abrolhos Islands, Australia |
| MU18NPI01 | 1 | d18O, Sr/Ca | (Murty et al., 2018b) | -8.67 | 115.51 | Nusa Penida, Indonesia (Lombok Strait) |
| NA09MAL01 | 4 | d18O | (Nakamura et al., 2009) | -3.2 | 40.1 | Malindi Marine Park, Kenya |
| PF04PBA01 | 4 | d18O | (Pfeiffer et al., 2004b) | -5.43 | 71.77 | Peros Banhos Atoll, Chagos Archipelago |
| PF19LAR01 | 1 | d18O, Sr/Ca | (Pfeiffer et al., 2019, 2004a) | -21 | 55 | St. Gilles Reef, La Reunion |
| RI10PBL01 | 4 | d18O | (Rixen et al., 2011) | 11.5 | 92.69 | Port Blair, Andaman Islands, India |
| ST13MAL01 | 1 | d18O, Sr/Ca | (Storz et al., 2013) | 4.29 | 72.98 | Rasdhoo Atoll, Maldives |
| WA17BAN01 | 2 | d18O, Sr/Ca | (Watanabe et al., 2017) | 23.5 | 58.75 | Bandar Khayran, Oman |
| ZI04IFR01 | 2 | d18O, Sr/Ca | (Zinke et al., 2004) | -23.15 | 43.58 | Ifaty Reef, Madagascar (Mozambique Channel) |
| ZI08MAY01 | 1 | d18O, Sr/Ca | (Zinke et al., 2008) | -12.65 | 45.1 | Mayotte (Comoro Archipelago) |
| ZI14HOU01 | 3 | d18O, Sr/Ca | (Zinke et al., 2014a) | -28.46 | 113.75 | Houtman Abrolhos Islands, Australia |
| ZI14IFR02 | 5 | d18O | (Zinke et al., 2014b, 2004) | -23.1573 | 43.5882 | Ifaty Reef, Madagascar |
| ZI14TUR01 | 5 | d18O | (Zinke et al., 2014b, 2004) | -23.3572 | 43.6195 | Tulear Reef, Madagascar |
| ZI15BUN01 | 7 | Sr/Ca | (Zinke et al., 2015) | -21.836 | 114.178 | Ningaloo Reef, Australia (Bundegi Reef) |
| ZI15CLE01 | 7 | Sr/Ca | (Zinke et al., 2015) | -17.26 | 119.26 | Rowley Shoals, Australia (Clerke Reef) |
| ZI15IMP01 | 7 | Sr/Ca | (Zinke et al., 2015) | -17.5369 | 118.974 | Rowley Shoals, Australia (Imperieuse Reef) |
| ZI15IMP02 | 7 | Sr/Ca | (Zinke et al., 2015) | -17.5196 | 118.969 | Rowley Shoals, Australia (Imperieuse Reef) |

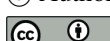

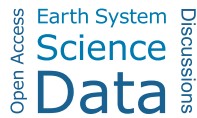

| | | | | | | |
|---|---|---|---|---|---|---|
| ZI15MER01 | 7 | Sr/Ca | (Zinke et al., 2015) | -17.1 | 119.6 | Rowley Shoals, Australia (Mermaid Reef) |
| ZI15TAN01 | 7 | Sr/Ca | (Zinke et al., 2015) | -21.893 | 113.963 | Ningaloo Reef, Australia (Tantabiddi Reef) |
| ZI16ROD01 | 6 | Sr/Ca | (Zinke et al., 2016) | -19.671 | 63.429 | Rodrigues, Republic of Mauritius (Totor Reef) |
| ZI16ROD02 | 6 | Sr/Ca | (Zinke et al., 2016) | -19.667 | 63.434 | Rodrigues, Republic of Mauritius (Cabri Reef) |
| *Red Sea* | | | | | | |
| BR19RED01 | 6 | Sr/Ca | (Bryan et al., 2019) | 19.89 | 39.96 | Canyon, Red Sea |
| DE16RED01 | 6 | Sr/Ca | (DeCarlo et al., 2016; Alpert et al., 2017) | 22.0314 | 38.8778 | Red Sea |
| FE18RUS01 | 1 | d18O, Sr/Ca | (Felis et al., 2018, 2000) | 27.8483 | 34.31 | Ras Umm Sidd, Egypt (Sinai Peninsula) |
| KL97DAH01 | 5 | d18O | (Klein et al., 1997; Ionita et al., 2014) | 15.7167 | 39.9 | Dur-Ghella Island, Eritrea (Dahlak Archipelago) |
| MU18RED01 | 6 | Sr/Ca | (Murty et al., 2018a) | 27.98 | 34.81 | Semicolon, Red Sea |
| MU18RED02 | 6 | Sr/Ca | (Murty et al., 2018a) | 25.58 | 36.55 | Popponesset, Red Sea |
| MU18RED03 | 6 | Sr/Ca | (Murty et al., 2018a) | 23.7 | 37.97 | Abu Galawa, Red Sea |
| MU18RED04 | 6 | Sr/Ca | (Murty et al., 2018a; Bryan et al., 2019) | 21.78 | 38.83 | Coral Gardens, Red Sea |

## 6 Author contributions

HRS, TF, KMC, and NJA directed the CoralHydro2k Project. RMW built and managed the CoralHydro2k database with supervision from HRS, technical assistance from MJF and assistance with conversion to LiPD format from NPM. All co-authors contributed to the design of the database, which includes the database format, record inclusion criteria, and metadata selection and standardization. Data curation efforts were led by HRS, RMW, BE, JAH, LDB and KHK, with RMW, HRS, TF, NJA, AKA, ARA, LDB, EPD, KLD, BE, MJF, NFG, JAH, KHK, HK, SAM, RDR, EVR, DS, SCS, and JZ populating either

data sets and/or metadata included in the CoralHydro2k database. Quality control efforts were led by RMW, HK, BE, MJF, and KHK, with assistance from AKA, LDB, ARA, EPD, KLD, TF, NFG, SAM, RDR, EVR, HRS, and JZ. Manuscript text was written by HRS and RMW, with significant contributions from TF, KHK, KLD, NFG, and JZ, and with all coauthors





providing edits and feedback throughout the process. RMW generated all figures for this paper with inputs from HRS, TF, KMC, NJA, ARA, KLD, BE, MJF, NFG, KHK, HK, DS, and SCS, and with all coauthors providing feedback on design and interpretation. HRS, RMW, and TF facilitated group meetings and workflows. TF, BE, and JAH organized the ICP13 in-person/online workshop held in Sydney, Australia in 2019.

## 7 Team List

The "CoralHydro2k Project Members" author group includes all named contributors as well as: Sarah Sr. Eggleston (Past Global Changes, 3012 Bern, Switzerland), Nicholas T. Hitt (School of Geography, Environment, and Earth Sciences, Victoria University of Wellington, Wellington, 6012, New Zealand), Belen Martrat (Department of Environmental Chemistry, Spanish Council for Scientific Research (CSIC), Institute of Environmental Assessment and Water Research (IDAEA), 08034 Barcelona, Spain), Helen V. McGregor (School of Earth and Environmental Sciences, University of Wollongong, Wollongong, 2522, Australia), Maria Rosabelle Ong (Lamont-Doherty Earth Observatory, Columbia University, Palisades, 10964, USA, and American Museum of Natural History, New York, 10021, USA), and Feng Zhu (School of Atmospheric Sciences, Nanjing University of Information Science & Technology, Nanjing, 211544, China).

## 8 Acknowledgements

CoralHydro2k is a contribution to Phases 3 and 4 of the PAGES 2k Network. We would like to thank PAGES IPO for providing logistical, technical, and financial support for community-driven projects such as CoralHydro2k. More specifically, we'd like to extend our gratitude to Sarah Eggleston, Belen Martrat, Angela Wade, and the PAGES2k coordinators for helping facilitate various aspects of this project over the last four years. Most importantly, we would like to thank the original data generators of each coral-based proxy record for making their data publicly available via the World Data Center PANGAEA, the NOAA NCEI World Data Service for Paleoclimatology, or other means, without whom this effort would not be possible.

We would like to thank all the researchers whose publicly archived data was included in the CoralHydro2k database (Appendix A). We would also like to thank the following researchers who provided data that were not previously archived or were archived in places other than NOAA or PANGAEA and are now included in CoralHydro2k and NOAA: Tianran Chen, Wenfeng Deng, Juan P. D'Olivo, Heitor Evangelista, Jennifer A. Flannery, Eberhard Gischler, Nathalie F. Goodkin, Y. Kawakubo, K. Halimeda Kilbourne, Braddock K. Linsley, Christopher R. Maupin, Hussein R. Sayani, Sujata A. Murty, David Storz, Takaaki K. Watanabe, Henry C. Wu, Hangfang Xiao.

We would also like to acknowledge software and data sources that made this project possible. Sea surface temperature data (ERSSTv5) was provided by the NOAA/OAR/ESRL Physical Sciences Laboratory in Boulder, Colorado, USA, from their website at https://psl.noaa.gov/data/gridded/data.noaa.ersst.v5.html. Salinity data (EN4.2.1) was provided by the Met Office Hadley Centre (www.metoffice.gov.uk/hadobs). All maps displayed in Figs. 1–7 were generated using the M_Map





MATLAB package found at www.eoas.ubc.ca/~rich/map.html. Lastly, we would like to thank NOAA NCEI and the LiPD team for facilitating archival and distribution of the CoralHydro2k database.

This work was partially funded by Georgia Tech's President's Undergraduate Research Award (PURA) and the Rutt Bridges Undergraduate Research Award to R. Walter. KLD partially funded by NSF Award #2102931 and Department of the Interior South Central Climate Adaptation Science Center Cooperative Agreement G19AC00086. NJA, BE and JAH
gratefully acknowledge funding from the Australian Research Council (FT160100029 and CE170100023).



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
