# Peer review of "The CoralHydro2k Database: a global, actively curated compilation of coral δ18O and Sr/Ca proxy records of tropical ocean hydrology and temperature for the Common Era"

_Earth System Science Data, 2022_

## Author Comment (AC1)

**RESPONSE TO REVIEWER #1**

We would like to thank Reviewer #1 for taking the time to review this manuscript and provide constructive feedback. We respond to each comment below, and where needed, outline changes made to the manuscript based on reviewer feedback or provide explanations as requested. We greatly appreciate the feedback and believe that it has substantially improved the revised manuscript.

Hussein Sayani and Rachel Walter,
On behalf of all authors

Dear authors, dear editor,

**Summary:** The PAGES CoralHydro2k Project team presents their database of paired coral δ18O and Sr/Ca proxy records in the manuscript in question. This is a major contribution for our understanding of tropical climatology over the Common Era and particularly the past 200 years. I have to emphasize that I cannot comment on the quality of the manuscript with respect to questions specific to coral research but I review it mainly with respect to general paleoclimatological and data publication aspects. I have a number of minor comments and questions on the manuscript, which, however, are not critical. I have two larger - though still not major - notes on the manuscript assets and the presentation.

From my point of view the manuscript can be quickly published.

Recommendation: I recommend publication after minor revisions.

*Larger notes:*

1. a. As far as I can see the document assets are not yet available at the already prepared persistent location (https://doi.org/10.25921/yp94-v135, last accessed 19. August 2022). This makes it hard to assess database version 1.0, which supposedly is described in the manuscript. I would prefer to have access to version 1.0 and not to have to fall back to version 0.5.4. Indeed, I don't think reviewers are clearly enough pointed to version 0.5.4 in the editorial interface but that may be more a comment towards Copernicus and not the authors. Version 0.5.4 is apparently a smaller subset of version 1.0. Therefore there is an obvious discrepancy between the data as described and the data that can be reviewed. I trust the authors and the editorial team at Copernicus that they ensure that (i) the data will be available at publication at

the given location and (ii) that version 1.0 will be as accessible with the available tools as version 0.5.4.

Version 0.5.4 is available at the Lipdverse (https://lipdverse.org/CoralHydro2k/current_version/, last access 19. August 2022).

Response: Version 0.5.4 is the official version 1.0 of the database. Our intention was to rename the database upon publication of the manuscript, however this was not clearly explained in the manuscript or in a note to reviewers. We apologize for the confusion. We have double checked version 0.5.4, and can confirm that it is a full version of the database and contains all the records described in the manuscript. We have also since uploaded correctly named versions of the database to each repository, and both the LiPDverse and NOAA archives can be fully accessed by reviewers. Links to example scripts and data submission forms are also now updated and point to the correct resource.

b. Going from the description in the manuscript, I am not sure what I should find at the given DOI. Neither can I verify that what is described at NOAA as downloadable data is identical with the manuscript description. NOAA NCEI states that there is a description of the file, an example file that may be the Matlab script mentioned in the manuscript but may also be something else, a pickle-file, a zip-file for the LiPD-files, an RData-file, a Matlab-data-file and a NCEI Direct Download, which at the moment does not point to a download but to the Google-form for submission of new records.

Response: The database and supporting files were submitted to NOAA, as file submission is required to create a page, however, the links are not active as we wanted the database and manuscript to pass peer-review before the data was publicly used. Reviewers were provided with a direct link to the same files on the LiPD website which is currently unlisted on the main page. We recognize that this may not have been communicated effectively and may have impeded our reviewers ability to fully vet the database and accompanying files. As such, we have requested that the links on the NOAA page be activated and files should now be accessible..

2. I am not convinced that the technicalities of accessing the data are sufficiently presented in the manuscript (its section 4) and in the supposedly accompanying Matlab-script. This is not least the case because I am uncertain how commonly the LiPD format is now used by colleagues.

The Matlab-script is, as far as I can find, not available so far and may or may not be included as a potential item at the database's NOAA NCEI access page.

I would welcome it if the authors provide more detailed documentation that quickly guides the potential user through loading the database, filtering it, and plotting an example series or even redoing one or more of the plots in the manuscript. I quote from the review guidelines: "The authors should point to suitable software or services for simple visualization and analysis, keeping in mind that neither the reviewer nor the casual "reader" will install or pay for it." While the authors point to the LiPD format and the associated tools and while the serializations can be accessed without any knowledge about LiPD, such a simple walk through may increase the later reuse and utilization of the database. An example of what I have in mind could be Nick McKay's (one of the manuscript's co-authors) tutorial for the geochronR package (e.g., https://nickmckay.github.io/GeoChronR/articles/Introduction.html, last accessed 19. August 2022).

Response: One of the larger goals of the 2k network is to archive all project data in a consistent, machine readable format. Both the PAGES 2k temperature and water isotope (iso2k)databases employed the LiPD format and have been widely used without issue. We opted to follow their lead and use a similar format. That being said, we agree with the reviewer that section 4 and supplemental materials can be improved to increase access to the database. To that extent, we have developed a Jupyter Notebook demonstration (https://github.com/CoralHydro2k/ch2k-notebooks) for accessing the database via Python, which is a free and easily accessible programming language, and have also created an example script for accessing the database using R. Together with Matlab, these languages cover >90% of our community. Resources for accessing all three versions of the database can be found on both the NOAA and LiPDverse archives. We have also reworded this section of the text to provide some more detail on how to query the database and what resources are available.

*Sidenotes:*

I did not test the access to the Matlab version. I did test the access to the Python and R serializations. I did test the access to the LiPD-files from within R. I only had a slightly more detailed look from R, which suggests that the data is accessible as described. I did not check the consistency of the appendix table.

Response: Thank you for checking both the R and Python serializations. Our goal is to make the database as accessible as possible by providing multiple formats and download locations.

**Minor**

Page 2, line 57ff: I am not sure this sentence is relevant for the topic of the manuscript. If it is from the authors' point of view, I nevertheless wonder if they really mean aspects of large-scale hydrology being tied to large-scale dynamics or if they mean more generally aspects of hydrology.

Response: This is a general statement about hydrology to provide motivation for compiling this database.

Page 3, line 89ff: The authors mention SISAL later in the paper, but I think the database is also relevant here.

Response: We have added references for SISAL which mentions other PAGES databases with CE records. The paragraph itself is primarily focused on coral records in previous databases, so the text itself remains unchanged.

Page 3, line 106ff: The paragraph includes the phrases "active curation" and "opportunities for future data collection". While both are indeed mentioned later, the phrasing here suggests more prominence for both than eventually realized.

Response: We have updated the text in section 4.3 to more clearly explain that new updates will include data submitted directly by record generators via the data submission form as well records sourced by the CoralHydro2k team from public archives.

Page 5, Line 153: I am surprised - and apparently didn't pay attention to Iso2k - by using only two digits of the publication year. In a sense it probably is a realistic perspective on the longevity of any data today but the philosophy may result in conflicts at some point.

Response: Potential for conflicts in record names certainly does exist in the very distant future. However, the record naming convention used for the Unique ID is a guide rather than a strict rule, and can be modified in future versions of the database should the need arise. We also do

not recommend using core names to search for records within the database, as there are several metadata fields that can be used for this purpose far more efficiently.

Tables generally: The authors clarify the meaning of "standardized" fields in the manuscript text, but I am not sure that the reader will get what is meant from the table captions alone.

Response: We've added the definition of standardized fields to the captions.

Table 2: This is minor but I think it may become important if more databases use comparable structures. The CoreID-variable has the fieldname "paleoData_ch2kCoreCode". May this ID better have a fieldname that is more directly interoperable with other LiPD IDs as it is of the same structure as an Iso2k ID - if I understand it correctly. What I mean is, if "paleoData_coreCode" or "paleoData_code" are potentially better fieldnames and other databases may, then, want to use the exact same fieldname.

Response: The database includes both "paleoData_ch2kCoreCode", a CoralHydro2k specific coreID, and "dataSetName", which is a more interoperable metadata field found in other LiPD databases. We now include a description for both in table 2. Future versions of the database may remove the paleoData_ch2kCoreCode.

Table 2: Similarly to the previous comment I wonder if "geo_secondarySiteName" is standard nomenclature for comparable types of data.

Response: The metadata field names follow PaCTS 1.0 and LiPD guidelines, as to our knowledge there are no universally accepted nomenclature standards for paleoclimate databases. Standardized location information fields are labeled more clearly, e.g. siteName, latitude, longitude, elevation, etc. The secondarySiteName field is not standardized and includes additional information, if it was provided by the authors in the original publication, that doesn't fit within the requirements of the other location metadata fields (e.g. colloquial names for a site, island names, reef names, etc). Table 2 now correctly indicates which fields are standardized and which ones are not.

Table 2: The authors call the paleoData_TSid a LiPD ID but it is also in the serializations. So, I am not sure whether LiPD in the manuscript refers to the data container or file format or

"vehicle" as the original paper calls it, or to the framework of structuring the data. Maybe it is not so much a LiPD ID but a "time series" or "record" ID.

Response: We have adjusted the wording to more accurately reflect field contents.

Table 2: I am a bit confused by the connection between the fieldname "paleoData_hasUncertainty" and the variable Error TSid. First, the fieldname for me suggests a logic variable or flag but not a TSid. For an ID, I would rather expect a fieldname like "paleoData_errorTSid" in agreement with the "paleoData_TSid". Second, I am of two minds if I agree that there should be a difference between Error TSid and TSid. Both are in the end TSids, both are generated the same way presumably. They serve different functions. I suggest to the authors to consider if Error TSid may better also be named TSid - but I myself tend right now to a "no".

Response: Renaming the fields is currently not possible; however, we have updated the definitions of this and associated field names in Table 2 to help users better understand their function.

Table 4: I personally would welcome standardization and quality control on the "Original data source" in an upcoming update on the database as well as inclusion of a persistent identifier (PID) for this Original source as an additional entry in the publication metadata. That is, an entry "Original data source PID" with variable "originalDataPID". Maybe that is even a major shortcoming of the database in its current state.

Response: In general, the first publication fields are the original or the first paper that published records from a core. Also included is a link to the data archive that each record was obtained from. We encourage users of each record to cite not just the original work, but all subsequent work as the CH2k database includes the most recent iteration of each record, which includes contributions from all studies listed in the metadata. Nonetheless, including an original data field is an intriguing idea and we'll leave it to future updates of the database to include this and adjust the variable name if needed.

Table 6: As a reader and a potential user - who likely would not get into the documentation first - I wonder if the entries "calibration_dataset" and "calibration_datasetRange" are clear enough transporting what they are or whether users may expect something different.

Response: We strongly encourage all potential users to consult the documentation provided prior to using the database. We tried to best balance name complexity with description accuracy, but we cannot account for all possible interpretations of each metadata field name when selecting them.

Section 3 generally but starting with section 3.2: Regarding the given significant correlations: a correlation of 0.13 may be significant but what really can we expect to learn from such a weak relation between proxy and variable of interest. Extending on that, are these very weak correlations significantly different from zero. To be clear, I do not expect the authors to answer by extra analysis but I think it should be commented on shortly.

Response: All correlations reported are significant and non-zero. Correlations are sensitive to the choice of SST dataset and gridbox used and age model uncertainties. Low correlations may also indicate the influence of other environmental factors on that proxy record (e.g. seawater d18O on d18O records). Our goal was to be inclusive and include as many coral records as possible, and the correlations are simply presented to highlight relationships and not necessarily their strength. It is left to the user to determine which records to use and how best to use them.

Section 3.2: again on the correlations: The authors state that a higher percentage of records is correlated significantly for bimonthly data than for annual data. How much of this potentially is due to seasonal signals? And if so, what does this imply for subsequent reconstructions, if the, e.g., annual cycle peaks dominate the correlation skill? Or has this basically no repercussions at all? The authors address this point in parts on page 18 in line 317. Thus, there also applies my question: is the seasonal cycle correlation a feature in records or may it even negatively affect reconstructions of interannual climate variability?

Response: Corals are one of the few archives that offer seasonally resolved records. The seasonal correlation is a feature for users looking to investigate or reconstruct seasonal variability, and this is something that the data assimilation community is particularly interested in. Coral proxy records from sites with a strong seasonal cycle in SST will typically exhibit a strong seasonal correlation with SST. The seasonal correlation is not reflective of the skill of these records at reconstructing interannual variability. As such, we have also provided interannual correlations with SST to highlight records with strong interannual variability. How these signals interact within a coral record and impact a reconstruction are beyond the scope

of a database manuscript and the subject of active research. We leave it to the end users of the database to determine how best to utilize the data.

Figure 5: I think it may be helpful to add more information on the filtering also to the caption. However, I also understand if, then, the caption becomes too lengthy.

Response: We've included as much information in the caption as possible while still keeping it concise and easy to understand. Any more detail would make the caption far too lengthy. More detailed methodology outlining how the filtering was performed and variance was calculated is provided in the 4th paragraph of section 3.2 where the figure is referenced.

Page 17, line 296: I am not sure that the authors use the term "mode of variability" in its commonly understood meaning here. Modes of variability usually - in my understanding - do not refer to frequency bands but to large scale features of climate variations.

Response: We have updated the wording to use "frequency" instead of "mode".

Page 18, line 303: Sentence: "Conversely …" Looking at Figure 5, my impression is that this statement is not correct in its absoluteness, but the authors have done more analysis than looking at the Figure, so my eyeballing may be wrong.

Response: We changed the wording to clarify that the highest interannual variance is observed in the Indo-pacific warm pool region.

Figures 6 and 7: Maybe the captions could benefit from some more details.

Response: We have updated the captions for Figure 6 and 7 to provide more clarity.

Page 20, line 366ff: The references for the sentence on "vital effects" are quite old. As I am not a coral-person I am curious: have there not been any updates on this topic?

Response: We used the original vital effects references in this sentence. While subsequent studies have cataloged, quantified, and verified vital effects, to our knowledge there are no concrete explanations to why these differences exist.

Page 21, line 380: I am surprised that the sentence singles out the impact of calibration. Isn't it more the impact of each step in the workflow that requires more work? Indeed this made me wonder if the coral community could do - or maybe they even already did it or are in the process of doing it - something like the tree ring community did for Büntgen et al. (2021, 10.1038/s41467-021-23627-6)?

Response: To our knowledge, no such work has been published for coral records. This may be in part due to the lack of a standardized, machine-readable archive of coral records. We're hoping that the CoralHydro2k database will help facilitate such efforts. The CoralHydro2k team is also actively exploring how best to calibrate coral proxies with environmental variables.

Page 21, line 399: I am not sure that "LiPD serialization" is clearly understandable, and that it is clear that the author's view on their data is that they provide (a) the database as in LiPD-formats and (b) a number of serializations of the database to serve different languages. In addition the following paragraph and list could be understood as meaning that these are the only possibilities to subset the data but - again unless I am mistaken - this list is not comprehensive.

Response: We have heavily reworded this section. We have removed references to the LiPD serialization, which we agree is confusing. We provide more information on how to search the database and indicate that these are just some but not all the ways in which the database can be queried. We also reference example scripts that are now archived with the database.

Page 21, line 398: Are D and TS correctly described as "variables" - not least as variable means something different in the data.

Response: To avoid confusion, we refer to D and TS as data containers.

Page 22, line 401: The authors write, the database can be searched. Naively one may assume that there are specialized tools for the database. It may help to clarify that, unless I am mistaken, the "searchability" basically means to use a coding language to reorganize the data.

Response: We've updated the text in section 4.2 to clarify that the databases need to be searched or filtered in MATLAB, R, or Python.

Page 22, line 418: It would be helpful if the MATLAB script was available already. It would also be a great service to the community, if further scripts or notebooks for other languages are provided in the future.

Response: The MATLAB scripts are now accessible on both NOAA and LiPDverse websites. We have also included a python demo and R script to increase accessibility of the database.

Page 22, line 425ff: "It is anticipated" is a rather weak statement. Does NOAA NCEI allow for such a change log and is CoralHydro2k striving to provide it?

Response: Given that the database is planned to be updated annually, and that few records are published each year, updating the files on NOAA NCEI and including a text-based change-log file is quite feasible. We have updated the wording of this statement.

Page 22, line 430: "If only a subset …": I disagree and I welcome if the authors change their message here. If any subset of the database is used each member of this subset should be referenced. Similarly, if any record is singled out, these records should be explicitly referenced. This ideally includes citations to a relevant publication and the record/dataset.

Response: We have updated the text to recommend that users cite the original and all relevant publications for each record used provided that it does not exceed reference limits of the target journal.

Page 23, line 432: I recommend that the authors also include a persistent identifier (PID) to the original public archive to foster FAIRness, reproducibility, provenance, and a culture of giving credit where credit is due.

Response: Version 1.0 of the database includes bibliographical information and DOIs for all publications associated with each record, with information for the original publication stored in the pub1 metadata fields. Moreover, a link to the original online public archive is also included in the metadata field "originalDataURL" as described on Table 4. Some of these links are DOIs, while others are either NOAA or PANGAEA links. We will work on converting all of these links to DOIs in a future release of the database.

Page 23, line 437: "improving the skill of future climate projections". I agree but I think this statement would benefit from a reference - or if the point is supposed to be made above already, then a reference and more emphasis are necessary there.

Response: This statement references the introduction, where we discuss the limitations of existing SST datasets and how they impact our ability to validate climate models.

Appendix table: I suggest that the authors include further information in this table: (a) a persistent identifier (PID), e.g., a DOI, for each record, (b) a data citation for each record, and (c) the DOIs for the publication. (c) may be unnecessary assuming the reference list is complete and (a) may also be obsolete if (b) is fulfilled and all data citations are in the reference list and include such a PID.

Response: Appendix A is a complete list of records in the database, and all relevant publications are cited in the table and in the reference list at the end of the manuscript. Moreover, all publication information, including links to the original archive are included in the database for each record.

Beyond that I did not check the consistency of this table.

Page 37, line 487: Acknowledgements: As former PAGES 2k coordinator I am unsure if CoralHydro2k received funding from PAGES within their Data Stewardship Scholarship, if so, I think this should be acknowledged and put into the funding information.

Response: CoralHydro2k did receive two Data Stewardship Scholarships, however, they were applied for and used to spearhead a separate initiative, which is the CoralHydro2k Seawater $\delta^{18}O$ database.

Software: If there is code to write or access the data structures, sharing it publically may foster wider adaptation of the CoralHydro2k database. This could also be referenced including "data"/code citations.

Response: A new demo and code are now archived with the database.

Finally, not so much a comment on the manuscript but on the chosen data format. As an R-user I still would welcome it if all the LiPD tools were available from CRAN and not only from Github. If I recall correctly, the LiPD-crew is pursuing this goal but I thought I might emphasize my wish once more here.

Response: While we agree that increased accessibility to LiPD is important, developing/providing LiPD tools is beyond the scope of CoralHydro2k and this review is not the best venue for this comment. We recommend addressing these wishes directly to the "LiPD-crew", either by email (linkedearth@gmail.com) or on Discourse (https://discourse.linked.earth/) for a more public forum.

**Technical**

Page 1, Line 39ff: I am not sure that the sentence "Most coral-based …" is clear on first reading. Maybe consider clarifying.

Response: We have reworded this sentence.

Page 3, Line 85ff: Again, I am not sure if the sentence "Whereas …" is clear for the reader. If I understand it correctly the main point is the contrast between success at sites and the limited assessment of larger scale signals. I think restructuring the sentence may clarify the point.

Response: This was a holdover from previous drafts. We have restructured the sentence to make our point clearer.

Page 4, Line 112: "is" and "make up". In a sense phase 3 is gone and PAGES 2k is now in phase 4 - and CoralHydro2k is still part of it. I wonder if it may be an idea to rephrase this to be more aligned to the current status. However, it isn't wrong as written, so may also stand.

Response: We have updated this to reflect that CoralHydro2k was part of Phase 3 and continues into Phase 4.

Pages 4, Line 121: Do Google Suite, Slack, and Zoom require references?

Response: No. Like the programming languages mentioned in the manuscript, we believe these are universally known software and do not need references.

Page 4, Line 114 and line 126: The authors mention the project goals in line 126 but I think they better fit in line 114.

Response: The goals of CoralHydro2k and this manuscript are already outlined in the section prior to this. As such, we have decided to preserve the current text.

Section 2.2: Is a reference to the FAIR principles already needed here?

Response: Section 2.2 discusses record selection. The FAIR principles deal with database structure and accessibility, and as such, is cited in section 4.3 which outlines database availability and format.

Tables in general: I do wonder if the clarity of the manuscript and the understanding of the tables would benefit from slightly more worded/detailed captions. Tables again: As nothing in Table 2 is italicized as far as I can see, I invite the authors to check the italicization in all tables.

Response: We have added missing italicization back to the tables. These seems to have been lost when transferring the manuscript to MS Word. We have also updated the captions for most tables to clarify what standardized terms are.

Metadata field names: Most fieldnames are structured as "word1_word2Word3" but the publication metadata in table 4 is simply written English. I think this should be aligned between different tables and if some changes have to happen to the data files, this should also be done.

Response: Thank you for catching this. The field name and variable column labels were accidentally flipped. We have updated Table 4.

Table 2: the description for paleoData_variableName has "will be" and "will have" and I wonder if the tense is correct.

Response: Fixed.

Figure 1: The Figure would become even clearer if there was a bit more white space between panels a and b but that certainly is a very minor point.

Response: This feedback is appreciated, but in the interest of time, we have instead focused on addressing the more major comments.

Page 17, line 286: The authors write of "significant discrepancies". Is this a tested significance or simply a figure of writing? If it is the latter, I suggest replacing the word.

Response: The significant discrepancies between SST datasets are extensively documented in the papers cited in this sentence. Analysis on the impacts of these discrepancies on coral proxy-SST calibrations and coral-based SST reconstructions are beyond the scope of this manuscript, and will instead be covered in an upcoming CoralHydro2k publication.

Page 22, line 415: I am not convinced that using LiPD follows from being guided by the FAIR principles.

Response: In the absence of any other alternatives, we have defaulted to using the LiPD format which is used by most of the other PAGES2k products and is built upon community-sourced metadata recommendations.

Page 22, line 420: I ask the authors to check that the form is correctly labeled as such on the repository website.
Response: The link to the data submission form has been fixed.

Citation: https://doi.org/10.5194/essd-2022-172-RC1

---

## Author Comment (AC2)

**RESPONSE TO REVIEWER #2**

We would like to thank Reviewer #2 for taking the time to review this manuscript and provide constructive feedback. We respond to each comment below, and where needed, outline changes made to the manuscript based on reviewer feedback or provide explanations as requested. We greatly appreciate the feedback and believe that it has substantially improved the revised manuscript.

Hussein Sayani and Rachel Walter,
On behalf of all authors

RC2: 'Comment on essd-2022-172', Anonymous Referee #2, 21 Sep 2022
Summary:

Walter et al. present the PAGES CoralHydro2k database of coral δ18O and Sr/Ca records for the Common Era. This work represents a large collaborative effort to collate and standardize coral proxy data into a machine-readable format. The publication of this data product will be of great benefit to the paleoclimate community, and I anticipate that the database will be used in many future studies. Overall, the manuscript is very well-written, and I recommend minor revisions.

The introduction clearly discusses the utility of coral geochemical proxies and their climate applications. The subsequent Methods and Key Characteristics sections are well-organized and easy to follow. I found it helpful that the authors provide the six tables with the metadata fields in the main text.

I also appreciate that the database includes shorter coral records that may have been excluded from other PAGES 2k data compilations, as these shorter records are still useful for reconstructing snapshots of tropical climate variability. As another beneficial outcome of this work, it is worth noting that an additional 27 previously unarchived records were submitted to the NOAA NCEI database. I am also glad to see that the authors provide a plan for updating the database annually.

My major suggestion is to expand the Usage Notes in Section 4 to provide a more comprehensive overview of how to access and query the Coral Hydro2k database. If there are space constraints, then additional details and specific examples could be provided in the Appendix. The authors mention that a MATLAB script will be provided, but I highly recommend also including example scripts written in Python and R to benefit a larger number of users. For example, a sample Jupyter Notebook that queries the database and performs some simple time series analysis would be very helpful.

Response: We have updated the usage notes section to clarify how the database can be searched. More importantly, we have also developed a new python demo and R example script, that will be archived along the MATLAB example script, to increase accessibility of the database.

The peer-review process could be a valuable opportunity for individuals not involved in CoralHydro2k to test the database and assess whether additional documentation and/or step-by-step guides would be beneficial. Similarly, to the other reviewer's comments, my major comments focus on the current availability of the database and its documentation. I suggest providing more comprehensive documentation in Section 4.2 of the main text that will help the reader/user successfully query the database. I was not able to access any of the files or the MATLAB example script using the links provided in the manuscript to test this out myself.

I recognize the authors are likely hesitant to make the database publicly available before the manuscript is accepted but given that this database is intended to be widely used in the paleoclimate community, it would be helpful to provide access to the reviewers. This would allow me to test the database, better connect the Methods section in the manuscript to the actual database, and importantly, provide more meaningful, constructive suggestions to improve the documentation and sample code.

Response: In addition to providing a python demo and R scripts with step by step instructions on accessing the database, we have now made the database and all accompanying documents fully accessible on both the NOAA archive and the LiPDverse page. The links on the NOAA page were previously offline, while reviewers were provided with an active link to the database on the LiPDverse website. However, we feel that this was not clearly communicated to reviewers. We hope that reviewers can now fully access all resources and test out the database.

Once the manuscript is published, I also suggest the authors utilize the benefits of open source and publish their code and documentation on GitHub. A community-based approach will allow the code to evolve and improve with time. For example, database users could post questions and submit issues if they experience any bugs in the code. As mentioned in the Summary, I also recommend the authors include example scripts for Python and R to help support a wider group of users.

Response: We have created GitHub repository (https://github.com/CoralHydro2k/ch2k-notebooks) with example code available in Python, R, and MATLAB.

Minor Comments and Questions:

It may be helpful to have a more obvious 'Submit New Data' button on the main NOAA NCEI CoralHydro2k landing page that directs people to the Google Form for submitting new records: https://www.ncei.noaa.gov/metadata/geoportal/rest/metadata/item/noaa-coral-35453/html.

Response: We have requested that the link be renamed as suggested, but there are limitations on the customizability of the NCEI landing page. We recommend visiting the NOAA Study Page, linked on the NCEI landing page.

The 'NCEI Direct Download' for the dataset files on the NOAA/WDS Paleoclimatology website currently goes to the Google Form for submitting new records. I am unsure if this is intentional or if this will need to be updated.

Response: We have requested an update to this link, however, as there are limitations on the customizability of the NCEI landing page. We are working to resolve this and recommend using the NOAA study page, linked on the NCEI landing page.

I recommend providing additional details about modern versus fossil corals in the introduction. Fossil corals are briefly mentioned in L240, but it is important to highlight that they are essential for reconstructing tropical climate variability during the earlier parts of the Common Era. The authors could briefly discuss and reference a few key studies that use fossil corals records to reconstruct tropical climate variability prior to the 1800s.

Response: We have added a few references as requested.

For the fossil coral records, does the database have a way to point to the modern coral dataset that was used to develop the SST calibration?

Response: This is unfortunately not included in this version of the database, however, it is something we will strongly consider for a future release. For now, the geographic metadata can

be used to identify modern corals and fossil corals from the same locations, and in most cases, the modern coral was likely used to generate an SST calibration for the fossil coral. Where available, the calibration information used by the original publication is also included in the calibration metadata fields (see Table 6).

How many of the original studies publish reconstructed δ18Osw values from paired Sr/Ca and δ18O measurements? In these instances, does the database include the original Sr/Ca and δ18O time series in addition to the δ18Osw time series?

Response: There are 19 studies in the database that published reconstructed δ18Osw values, and each δ18Osw record is accompanied by their original Sr/Ca and δ18O time series under the same unique identifier.

None of the metadata field names are italicized in Table 2. I recommend double checking the fields and adjusting the italicized text accordingly.

Response: Thank you for flagging this. The italicization was lost when transferring the manuscript to MS word prior to submission. We have corrected this and updated the definitions in Table 2.

I recommend noting in L160-162 that the unstructured metadata fields are those not italicized in Tables 2-6.

Response: We have added a note as requested.

I am unsure what the 'paleoData_TSid' metadata field means in practice. Additional details in the description would be helpful.

Response: We have updated the descriptions for this and other related metadata fields in table 2 to better explain their purpose.

If any subset of the CoralHydro2k database is used (not just a small subset as discussed in L430), I think it would be beneficial to cite all the relevant original publications especially if this does not cause the new study to exceed the reference limit for the target journal.

Including the original DOIs for each database entry in Appendix A would facilitate this process.

Response: We agree and have updated the text in this section as requested. Appendix A includes all relevant citations for each record and their bibliographies, including DOIs, are included on the reference list.

Citation: https://doi.org/10.5194/essd-2022-172-RC2

---

## Referee Report (RR1)

**Review:**

The PAGES CoralHydro2k database of coral $\delta^{18}O$ and Sr/Ca records for the Common Era will be a welcome addition to the paleoclimate community. The writing remains excellent, and the figures and tables are high-quality. I recommend publication.

The authors thoroughly addressed all my comments. They did this by clarifying sections of the main text, revising the descriptions for metadata fields, and expanding the amount of sample code to include examples in MATLAB, Python, and R. I greatly appreciate the authors' efforts to develop a GitHub repository and provide examples in all three programming languages because this will make the database accessible to more users.

I was also pleased that the authors updated the usage notes in Section 4.2 to clarify how to search the database. I also appreciated the revised text in Section 4.4 that encourages database users to cite the original publications for the coral records whenever possible.

I was able to access the NOAA Study Page and successfully download the database. I also cloned the CoralHydro2k MATLAB and Python GitHub repositories and successfully tested all the MATLAB and Python example scripts. The example code is helpful and easy to follow.

**Minor note about accessing the database:**
In the point-by-point response to reviews the authors note that there are limitations on the customizability of the NCEI landing page for the data DOI, and therefore, recommend visiting the NOAA Study Page to find the 'Submit New Data' link and access the code repository. This was a helpful comment, and I suggest including this recommendation in Section 4.3 of the manuscript. Another option is to include a direct link to the main study page (https://www.ncei.noaa.gov/access/paleo-search/study/35453) in addition to the data DOI. I also think it would be beneficial to include a direct link to the CoralHydro2k GitHub repository in the manuscript.

---

## Author Response (AR2)

**RESPONSE TO REVIEWER #1**

We would like to thank Reviewer #1 for reviewing the revised manuscript, testing the database and the example scripts, and provided constructive feedback. We followed the reviewer's recommendation to include links to the NOAA Study Page and GitHub repository as we agree that this would greatly improve accessibility to the CoralHydro2k database.

Hussein Sayani and Rachel Walter,
On behalf of all authors

**Review #1**

The PAGES CoralHydro2k database of coral $\delta^{18}O$ and Sr/Ca records for the Common Era will be a welcome addition to the paleoclimate community. The writing remains excellent, and the figures and tables are high-quality. I recommend publication.

The authors thoroughly addressed all my comments. They did this by clarifying sections of the main text, revising the descriptions for metadata fields, and expanding the amount of sample code to include examples in MATLAB, Python, and R. I greatly appreciate the authors' efforts to develop a GitHub repository and provide examples in all three programming languages because this will make the database accessible to more users.

I was also pleased that the authors updated the usage notes in Section 4.2 to clarify how to search the database. I also appreciated the revised text in Section 4.4 that encourages database users to cite the original publications for the coral records whenever possible.

I was able to access the NOAA Study Page and successfully download the database. I also cloned the CoralHydro2k MATLAB and Python GitHub repositories and successfully tested all the MATLAB and Python example scripts. The example code is helpful and easy to follow.

**Minor note about accessing the database:**

In the point-by-point response to reviews the authors note that there are limitations on the customizability of the NCEI landing page for the data DOI, and therefore, recommend visiting the NOAA Study Page to find the 'Submit New Data' link and access the code repository. This was a helpful comment, and I suggest including this recommendation in Section 4.3 of the manuscript. Another option is to include a direct link to the main study page (https://www.ncei.noaa.gov/access/paleo-search/study/35453) in addition to the data DOI. I also think it would be beneficial to include a direct link to the CoralHydro2k GitHub repository in the manuscript.

*Response: We have followed this recommendation and added links to the NOAA Study Page and GitHub in section 4.3. Currently, there is no way to edit the NCEI landing page as it is automatically generated. We are hoping that the landing page will be improved by NOAA in the near future. In the meantime, we are exploring the possibility of having the DOI point directly to the NOAA Study Page. If this is possible, it likely will not be completed at the time of this submission. We will link to the NOAA Study Page and direct users there and adjust these instructions in the final proof if warranted.*

**RESPONSE TO REVIEWER #2**

We would like to thank Reviewer #2 for reviewing the revised manuscript and providing constructive feedback. We respond to each comment below. To addressed the main feedback regarding database accessibility, we have followed Reviewer #1's suggestion for adding a direct link to the NOAA Study Page in the manuscript and directing readers there as the NCEI Landing Page cannot be edited at this time. We have also followed Reviewer #2's suggestion of archiving the original code for version 1.0 of the database on the NOAA NCEI repository. This code will be updated, if needed, and archived with each new release of the database.

Hussein Sayani and Rachel Walter,
On behalf of all authors

**Review #2**

Dear authors, dear editor,

I would like to thank the authors for so thoroughly addressing the referees' comments.

I focussed my new assessment on the question, whether I feel that the authors addressed prior comments regarding availability and usability of the data and the answer is yes.

Due to personal constraints, I did not check the manuscript in detail.

I have two small points to make.

1. I would just like to nitpick that putting code on Github is not a persistent way to store it. To ensure persistence, the authors could, for example, provide a current versioned copy in their NCEI repository.
*Response: We have archived the original code for version 1.0 of the database on the NCEI repository. We will update this code and archive as needed with future releases of the database.*

2. My other point is a technicality that probably requires the authors to get in contact with NOAA/WDS. While the paleo-search entry for the CoralHydro2k database (https://www.ncei.noaa.gov/access/paleo-search/study/35453) appears to correctly label all the entries, the geoportal entry (https://www.ncei.noaa.gov/metadata/geoportal/rest/metadata/item/noaa-coral-35453/html), which is connected to the DOI, still appears to contain potentially wrong links. That is, the form for new entries as well as the Github repository are both labeled as "NCEI Direct Download".
*Response: This is unfortunately a limitation of the current NCEI Landing Page, which is automatically generated and cannot be edited. Following the recommendation of reviewer #1, we have added a link to the NOAA Study Page to the manuscript and will direct readers there. We are also exploring some potential solutions with NOAA. If these are implemented before the final proof stage, we will adjust the text as needed.*

Technical notes:

Table 6 also includes the italicization-comment but has no italicized elements. Could the comment be replaced by something like: "There are no standardized elements in this table."

*Response: Thank you for catching this. We've followed your suggested and added a note that indicates there are no standardized metadata fields on this table.*

Page 22, line 26: Should the first part of this line be part of the list of items above?
*Response: Thank you for catching this. We've fixed the formatting issue.*

In case of questions, I invite the editor and the authors to contact me.

Best regards

Oliver Bothe